# Spatial control of translation repression and polarized growth by conserved NDR kinase Orb6 and RNA-binding protein Sts5

Illyce Nuñez[1], Marbelys Rodriguez Pino[1], David J Wiley[1], Maitreyi E Das[2], Chuan Chen[1], Tetsuya Goshima[3], Kazunori Kume[4], Dai Hirata[4], Takashi Toda[4,5], Fulvia Verde[1,6]*

[1]Molecular and Cellular Pharmacology, University of Miami School of Medicine, Miami, United States; [2]Department of Biochemistry and Cellular and Molecular Biology, The University of Tennessee, Knoxville, United States; [3]National Research Institute of Brewing, Higashi-Hiroshima, Japan; [4]Department of Molecular Biotechnology, Graduate School of Advanced Sciences of Matter, Hiroshima University, Higashi-Hiroshima, Japan; [5]The Francis Crick Institute, Lincoln's Inn Fields Laboratory, London, United Kingdom; [6]Marine Biological Laboratory, Woods Hole, United States

**Abstract** RNA-binding proteins contribute to the formation of ribonucleoprotein (RNP) granules by phase transition, but regulatory mechanisms are not fully understood. Conserved fission yeast NDR (Nuclear Dbf2-Related) kinase Orb6 governs cell morphogenesis in part by spatially controlling Cdc42 GTPase. Here we describe a novel, independent function for Orb6 kinase in negatively regulating the recruitment of RNA-binding protein Sts5 into RNPs to promote polarized cell growth. We find that Orb6 kinase inhibits Sts5 recruitment into granules, its association with processing (P) bodies, and degradation of Sts5-bound mRNAs by promoting Sts5 interaction with 14-3-3 protein Rad24. Many Sts5-bound mRNAs encode essential factors for polarized cell growth, and Orb6 kinase spatially and temporally controls the extent of Sts5 granule formation. Disruption of this control system affects cell morphology and alters the pattern of polarized cell growth, revealing a role for Orb6 kinase in the spatial control of translational repression that enables normal cell morphogenesis.

*For correspondence: fverde@miami.edu

## Introduction

Many cellular processes, such as cell morphogenesis, migration, and asymmetric cell division, require eukaryotic cells to alter polarity and growth patterns (*Lalli, 2014*; *Tahirovic and Bradke, 2009*; *Woodham and Machesky, 2014; Knoblich, 2008*). Understanding the conserved mechanisms by which cells tune polarized cell growth has implications for studies of neuronal cell morphogenesis, neurodegenerative diseases, stem cell differentiation, and cancer (*Yoshimura et al., 2006*; *Tahirovic and Bradke, 2009*; *Yamashita et al., 2010*; *Tanos and Rodriguez-Boulan, 2008*). The fission yeast *Schizosaccharomyces pombe* is an excellent model system to study cell morphogenesis and growth because cells have a defined cylindrical shape that enables straightforward evaluation of changes in growth and polarity. Under exponential growth conditions, fission yeast cells display a paradigmatic pattern of cell growth, growing in a monopolar fashion during early interphase and activating bipolar growth at the new cell tip once a minimal cell length has been achieved

**eLife digest** Living cells can grow to adopt a range of different shapes. For example, fission yeast cells maintain their rod-like shape by growing from their ends and then splitting in the middle to produce two new cells of an equal size. Like many other cells, fission yeast often responds to shortages of nutrients or other environmental stressors by growing more slowly or stopping its growth altogether. One way that this stress response is achieved is by preventing certain growth-promoting proteins from being made by storing or degrading the RNA molecules that are needed to make these proteins.

Fission yeast uses an enzyme called Orb6 to control its growth and its overall shape. This enzyme is a kinase, meaning that it adds phosphate groups on to other proteins. Orb6 controls cell growth in part by defining the scope of action of an important growth control factor, Cdc42. This is done by preventing the localization of a molecule that activates Cdc42 at the wrong places of the cell membrane, such as, for example, the cell sides.

Now, Nuñez et al. show that the Orb6 enzyme also controls growth via a completely separate mechanism. Orb6 prevents an RNA-binding protein called Sts5 from being recruited into clusters of RNA molecules and proteins called granules, and directs Sts5 to interact with another protein called Rad24 instead.

Since the RNA molecules in the granules tend to end up being degraded, preventing Sts5 from being recruited to the granules protects the RNA molecules that bind to Sts5. Many Sts5-bound RNAs encode proteins required for cell growth, and in this manner Orb6 promotes the production of these Sts5-controlled proteins to encourage normal cell growth.

In the future, Nuñez et al. would like to determine how Orb6 recognizes and responds to environmental signals to control cell growth. Further studies could also explore how Sts5 and Orb6 kinase affect the assembly of RNA-protein particles that have been implicated in diseases in humans.

(*Mitchison and Nurse, 1985*). Further, *S. pombe* displays a distinct morphological response to nutrient deprivation, which causes cells to divide at a shorter cell length and grow in a monopolar fashion (*Su et al., 1996*; *Yanagida, 2009*; *Yanagida et al., 2011*).

The NDR (Nuclear Dbf2-Related) kinase family with roles in cell morphogenesis, cell growth and proliferation, mitosis, and development, is highly conserved in cells ranging from yeast to mammalian neurons (*Verde et al., 1995*; *Verde et al., 1998*; *Zinn, 2004*; *Hergovich et al., 2006*). In humans, this subset of the AGC kinase group comprises NDR1 and NDR2 and the closely related kinases LATS1 (large tumor suppressor 1) and LATS2 (*Hergovich et al., 2006*), which function downstream of the MST/Hippo kinases (*Meng et al., 2016*). While LATS1 and LATS2 kinases are central to the Hippo pathway that plays a role in organ size and tumor suppression, dysregulation of NDR kinases has been implicated in cancers such as progressive ductal cell carcinoma, melanoma, non–small-cell lung cancer, and T-cell lymphoma (*Adeyinka et al., 2002*; *Millward et al., 1998*; *Hauschild et al., 1999*; *Ross et al., 2000*; *Cornils et al., 2010*). In addition to their link to cancer, NDR kinases function also in neuronal growth and differentiation, dendritic branching, and dendritic tiling, and have been implicated in memory and fear conditioning (*Emoto et al., 2004*; *Zallen et al., 2000*; *Koike-Kumagai et al., 2009*; *Stork et al., 2004*). Recent work has shown that mammalian NDR1 and NDR2 promote polarity in neurons upstream of the polarity protein Par3 (*Yang et al., 2014*). However, the mechanisms by which NDR kinases control cell growth and polarity are not fully understood. The fission yeast NDR kinase Orb6 is a central component of the conserved morphogenesis (MOR) regulatory network (*Hergovich et al., 2006*). We previously showed that NDR kinase Orb6 has a role in the establishment of cell polarity and the control of polarized cell growth (*Verde et al., 1995*; *Verde et al., 1998*). Orb6 kinase regulates cell polarity, in part, by spatially controlling conserved GTPase Cdc42 (*Das et al., 2009*), via inhibitory phosphorylation of Cdc42 guanine exchange factor (GEF) Gef1 (*Das et al., 2015*).

Here, we describe a novel role for Orb6 kinase, genetically separable from its control of the Cdc42 pathway, in promoting polarized cell growth by inhibiting translational repression. Translational repression, carried out in part by the assembly of cytoplasmic granules of ribonucleoprotein

particles (RNPs), is a quick and reversible cellular strategy for inhibiting cell growth in response to stress, such as nutritional deprivation, oxidative stress, or osmotic stress (*Coller and Parker, 2005*; *Decker and Parker, 2012*; *Kedersha et al., 2005*; *Jud et al., 2008*). P-bodies, stress granules, and other RNPs such as neuronal transport granules and germ granules play important roles in mRNA regulation with implications for human diseases such as ALS, frontotemporal lobar degeneration, and viral infection (*Ramaswami et al., 2013*; *Chahar et al., 2013*). P-bodies in particular contain mRNA decay machinery and serve as sites of storage or degradation for mRNAs during times of cellular stress (*Decker and Parker, 2012*). In this work, we describe a novel mechanism whereby NDR kinase Orb6 negatively regulates the recruitment of mRNA-binding protein Sts5 into RNP particles and Sts5 localization to P-bodies at least in part by promoting Sts5 interaction with 14-3-3 protein Rad24. This mechanism of control prevents the degradation of mRNAs encoding proteins important for polarized cell growth and cell morphogenesis during exponential cell growth, and promotes morphological adaptation during nutritional stress.

## Results

### Loss of RNA-binding protein Sts5 suppresses the cell viability defects of *orb6* mutants

We observed that loss of Orb6 kinase activity by chemical inhibition of analog-sensitive Orb6-as2 kinase by the ATP analogue 1-NA-PP1 leads to cell separation defects (*Figure 1A,c; B*) and slow growth, in addition to polarity defects (*Das et al., 2009*; *Das et al., 2015*). By complementation screening of the *orb6-as2* allele with mutants of other *orb* genes (*Snell and Nurse, 1994*; *Verde et al., 1995*), we found that *sts5* mutants (allelic to *orb4*; see *Figure 1—figure supplement 1A*) suppress the cell-separation defect associated with chemical inhibition of Orb6-as2 kinase (*Figure 1A,d*; *Figure 1B*; *Verde et al., 1995*) as compared to control cells (*Figure 1A,c*; *Figure 1B*). *sts5* encodes an mRNA-binding protein with significant sequence homology to Ribonuclease II (RNB)–domain and Ribonuclease R–domain proteins (*Toda et al., 1996*; *Jansen et al., 2009*). Closest homologues of Sts5 include *S. cerevisiae* Ssd1 (*Jansen et al., 2009*), *S. pombe* Dis3L2, and the human exonuclease Dis3L2, which has been associated with diseases such as Perlman syndrome and Wilm's tumor, as well as Rrp44/Dis3 (*Figure 1C*) (*Malecki et al., 2013*; *Robinson et al., 2015*; *Lv et al., 2015*; *Astuti et al., 2012*). Sts5 and Dis3L2 contain conserved domains (cold shock domains CSD1 and CSD2 and the S1 domain) that mediate interaction with the single-stranded RNA substrate (*Faehnle et al., 2014*). However, both Sts5 and Dis3L2 lack the PIN domain and CR3 signature amino acids that are implicated in the association of Rrp44/Dis3 to the exosome (Indicated by • in *Figure 1C*) (*Malecki et al., 2013*; *Schaeffer et al., 2012*; *Makino et al., 2013*; *Robinson et al., 2015*). Furthermore, Sts5 lacks conserved amino acids involved in RNA hydrolysis (marked by ▲ in *Figure 1C*), indicating that it is unlikely to have exonuclease activity (*Uesono et al., 1997*; *Jansen et al., 2009*).

Next, we investigated whether *sts5Δ* suppresses the loss of viability observed with temperature-sensitive *orb6* mutants and mutants of other components of the Orb6 pathway (*Verde et al., 1995*; *Verde et al., 1998*). Orb6 kinase belongs to the morphogenesis (MOR) network, which includes Nak1 kinase and its binding partner Pmo25, the scaffolding protein Mor2, and the Orb6 binding partner Mob2 (*Figure 1D*) (*Kanai et al., 2005*; *Hou et al., 2003*). As with *orb6* mutants, temperature-sensitive MOR mutants exhibit a loss of viability at the restrictive temperature. In a spot growth assay, we found that the *sts5-276* mutation (which truncates the *sts5* gene to a short 53-bp fragment; see *Figure 1—figure supplement 1A*) suppresses the temperature-sensitive growth defect associated with *orb6* mutation, as well as the growth defects of other MOR network mutants (*Figure 1E*).

Finally, we tested the idea that the function of *sts5* deletion in suppressing the loss of viability of *orb6* mutant cells is independent of Cdc42 GTPase. We found that *sts5* deletion does not suppress the cell rounding induced by prolonged Orb6 kinase inhibition (*Figure 1—figure supplement 1B*). Further, mislocalization of the Cdc42 reporter CRIB-GFP in *orb6* mutants (*Tatebe et al., 2008*; *Hoffman and Cerione, 2000*), a hallmark of Orb6 kinase inhibition (*Das et al., 2009*), is not suppressed by loss of *sts5* (*Figure 1—figure supplement 1C and 1D*). Conversely, deletion of the Cdc42 GEF *gef1* does not suppress the loss of viability of *orb6* mutants (*Figure 1—figure*

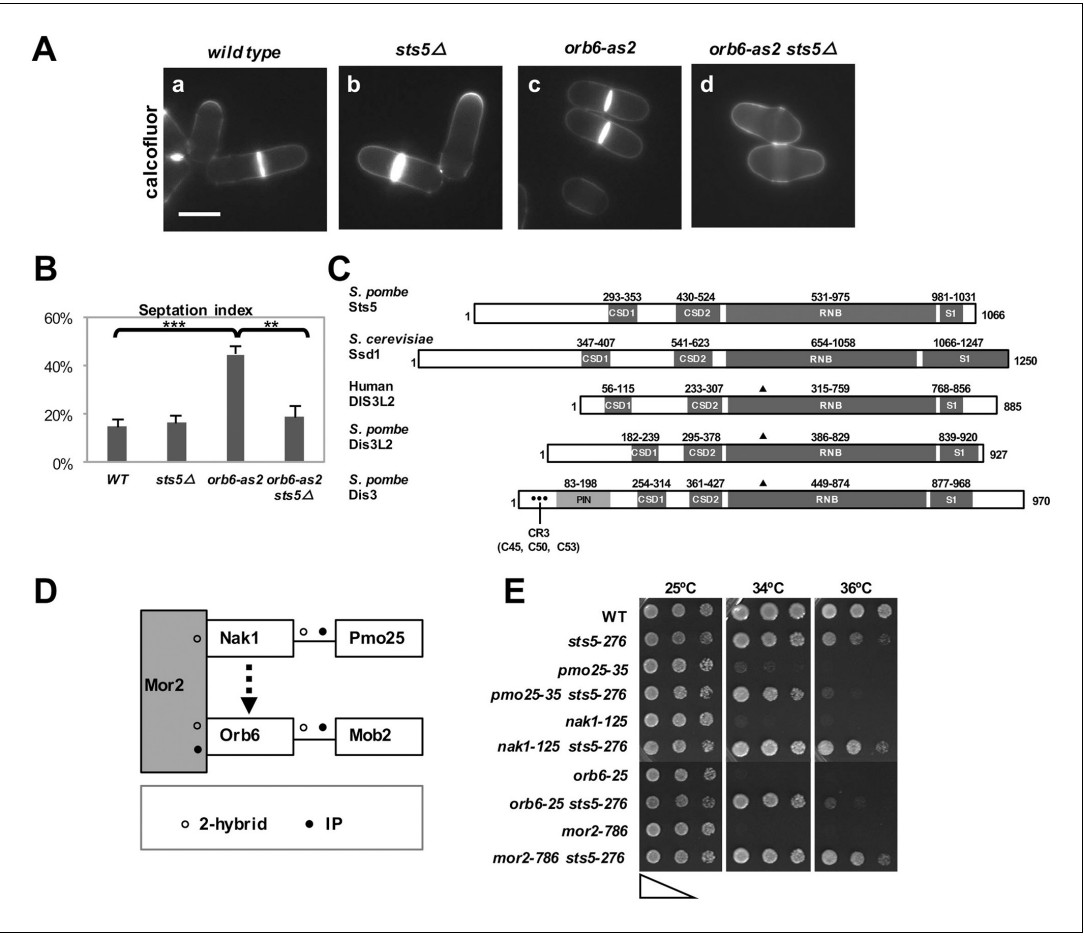

**Figure 1.** Loss of RNA-binding protein Sts5 suppresses the cell viability defects of *orb6* mutants. (**A**) Deletion of *sts5* suppresses the cell separation phenotype of analog-sensitive *orb6-as2* mutants. (**a**) wild-type, (**b**) *sts5Δ*, (**c**) *orb6-as2* and (**d**) *orb6-as2 sts5Δ* mutants treated with 50 μM 1-NA-PP1 inhibitor for 2 hr at 32°C. Bar = 5 μm. (**B**) Septation index quantification of cells in experiment shown in **A** based on 3 independent experiments (N>295 per strain). Orb6-as2 cells exhibit a significantly higher septation index as compared to control cells ($P = 0.0004$) and as compared to *orb6-as2 sts5Δ* double mutants ($P = 0.0011$). P values were determined using analysis of variance (ANOVA) with SPSS statistics package 22.0, followed by Tukey's HSD test. Error bars indicate SD. (**C**) Sts5 protein sequence includes the RNB domain (a.a. 531–975), with homology to the catalytic domain of *E. coli* ribonuclease II, three conserved OB-fold domains that promote interaction with RNA, the CSD1 (a.a. 293–353), CSD2 (a.a. 430–524), and S1 (a.a. 981–1031) domains. Sts5 is related to the exoribonuclease Dis3L2 that is conserved from *S. pombe* to humans. ▲ indicates 3 catalytic residues in RNB domain. • indicates CR3 motif residues for exosome targeting. (**D**) Interactions between MOR network proteins. Mor2 serves as a scaffold that enables activation of Mob2-bound Orb6 by the Nak1-Pmo25 complex (○ indicates 2-hybrid interaction; • indicates IP interaction). (**E**) *sts5-276* mutation suppresses the temperature-sensitive growth of MOR mutants. The indicated cells were spotted on YPD solid medium (approximately $5 \times 10^4$ cells in the left spots for each plate and then diluted 4-fold in each subsequent spot) and incubated at 25°C, 34°C, and 36°C for 3 days.

The following figure supplements are available for figure 1:

**Figure supplement 1.** Loss of RNA-binding protein Sts5 does not suppresses the polarity defects observed upon Orb6-as2 kinase inhibition.

**Figure supplement 2.** Deletion of *gef1* or *dis3L2* does not suppress the growth defect observed upon Orb6-as2 kinase inhibition.

*supplement 2A*), while it suppresses the polarity defect associated with Orb6 kinase inhibition (*Das et al., 2009*).

Together, these results suggest that Sts5 mediates the cell separation defects and loss of viability observed in *orb6* mutants. Further, this novel role of Orb6 kinase in growth control is genetically separable from its previously established function in the spatial control of Cdc42 GTPase.

## Sts5 proteins are recruited into cytoplasmic puncta during mitosis and during nutritional starvation

We used fluorescence microscopy to study the localization of Sts5-3xGFP. We found that during exponential growth Sts5-3xGFP localization is mostly diffuse in the cytoplasm during interphase (I) (*Figure 2A*). Sts5 coalesces into cytoplasmic puncta during mitosis as previously reported (*Vaggi et al., 2012*).

We found that upon nutrient starvation Sts5-3xGFP proteins rapidly coalesce into distinct, larger cytoplasmic puncta. Many of these puncta colocalize with P-body (Processing body) marker Dcp1-mCherry, a component of the mRNA decapping complex (*Wang et al., 2013*). Sts5 localization to the P-bodies is particularly strong during glucose deprivation (*Figure 2B,b,h*) and occurs also during nitrogen starvation, although puncta appear smaller and co-localization of Sts5 with Dcp1 is partial (*Figure 2B,c,i*). Conversely, Sts5 recruitment into puncta during mitosis occurs in the absence of substantial P-body formation, as visualized with P-body marker Dcp1-mCherry (*Figure 2A*). Consistent with these findings, Sts5 contains a region predicted to be intrinsically disordered in the first 301 amino acids of the Sts5 protein, a feature shared by many proteins that undergo assembly into RNP particles (*Lin et al., 2015*; *Patel et al., 2015*; *Elbaum-Garfinkle et al., 2015*; *Wang et al., 2014*; *Kato et al., 2012*; *Han et al., 2012*) (*Figure 2—figure supplement 1*). These results indicate that Sts5 proteins can organize in cytoplasmic puncta during mitosis and in response to nutritional stress. Furthermore, these puncta co-localize with P-bodies following glucose or nitrogen starvation.

## Loss of Sts5 leads to increased levels of mRNAs involved in growth control and bipolar growth activation

It is possible that Sts5 modulates cell growth by controlling the levels of the mRNAs it binds. To investigate the functions of Sts5-regulated transcripts, we first used microarray analysis to identify mRNAs that are elevated (at least 1.9-fold) in *sts5Δ* mutants as compared to wild-type cells, under exponential growth conditions (See *Figure 3—source data 1*). This analysis identified 140 mRNAs, and showed significant overrepresentation of genes with roles in polarized cell growth, adhesion and cell wall biogenesis by gene ontology enrichment analysis (*Figure 3*, *) (See *Figure 3—source data 1*). Remarkably, we identified *ssp1*, *cmk2*, *tea5/ppk2*, *ksg1* and *lkh1*, which encode protein kinases with a role in the activation of bipolar cell growth (*Figure 3*; *Figure 3—source data 1*) (*Koyano et al., 2010*). Intriguingly, these mRNAs encoding polarity regulators all contain a consensus sequence in their 5'UTR which functions as a potential recognition sequence for targets of the *S. cerevisiae* homolog of Sts5, Ssd1 (*Figure 3*, †) (*Hogan et al., 2008*) (*Wanless et al., 2014*). The fully categorized list of mRNAs identified in the microarray analysis includes mRNAs involved in cell wall biogenesis and secretion, cytoskeletal organization, nutrient transport, and meiosis (See *Figure 3—source data 1*).

The mRNAs encoding proteins with known roles in polarized cell growth were selected for further analysis by qPCR, which confirmed that *sts5Δ* cells exhibit increased levels of *ssp1*, *cmk2*, *tea5/ppk2*, *lkh1*, *efc25*, and *psu1* mRNA, genes with diverse functions in cell morphogenesis (*Figure 4A*) (*Figure 3*; *Figure 3—source data 1*). Further, we determined that Sts5-3xGFP protein co-purifies with *efc25*, *ssp1*, and *psu1* mRNAs (*Figure 4B*). Consistent with a functional interaction with *sts5* and *orb6*, *ssp1* was previously identified as an extragenic suppressor of *sts5* mutants (*Matsusaka et al., 1995*), while *psu1* functions as a multicopy suppressor of *orb6* mutants (*Figure 4—figure supplement 1*).

We chose to measure protein levels of HA-tagged Ssp1 and Myc-tagged Efc25 to gauge how Sts5 regulation of *ssp1* and *efc25* mRNAs affected the levels of Ssp1 and Efc25 proteins. We determined that Ssp1 protein levels significantly increase in *sts5Δ* mutants, as compared to controls, when cells are exposed to higher temperatures (36°C; *Figure 4C,D*). Interestingly, *sts5Δ* cells display a temperature-sensitive morphological phenotype, growing over a wider area of the cell surface and

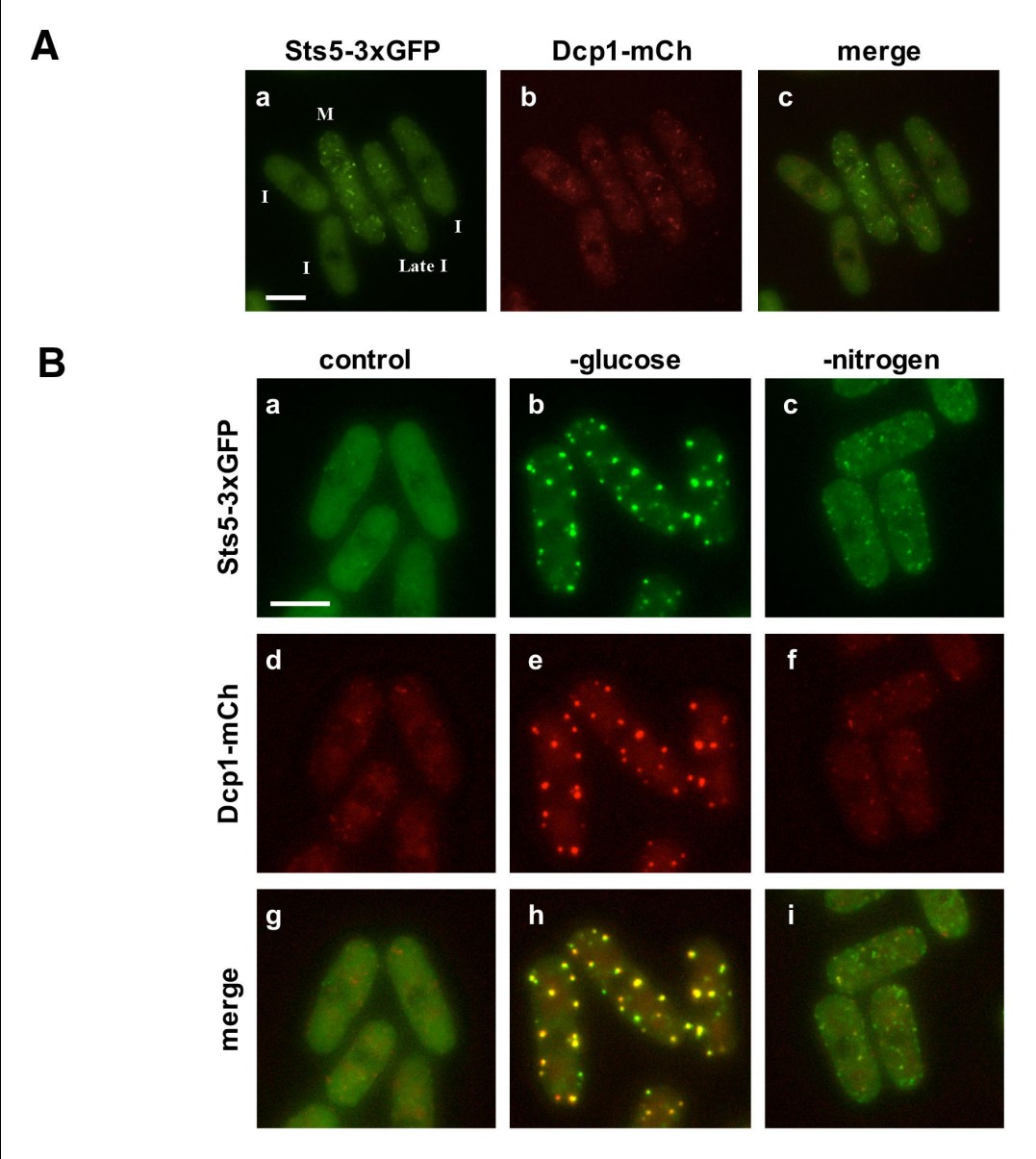

**Figure 2.** Sts5 proteins assemble into puncta during mitosis and during nutritional starvation. (**A**) Sts5-3xGFP proteins coalesce into cytoplasmic particles in cells undergoing mitosis (M) but appear mostly diffuse in the cytoplasm of growing interphase (I) cells (**a, c**). (**b**) P-body formation, as visualized by P-body marker Dcp1-mCherry is not induced in mitotic cells. Bar = 5 µm. (**B**) Sts5-3xGFP proteins are recruited and colocalize with the P-body marker Dcp1-mCherry upon growth for 1 hr in minimal medium minus glucose (**b, e, h**). Sts5-3xGFP recruitment and colocalization with Dcp1-mCherry in P-bodies also occurs upon 1 hr of growth in minimal medium minus nitrogen (**c, f, i**). Sts5-3xGFP recruitment was observed as early as 15 min after transfer to glucose- or nitrogen-depleted medium. Images are deconvolved projections from 12 Z-stacks separated by a step size of 0.3 µm. Experiment was performed using prototrophic strain FV2267. Bar = 5 µm.

The following figure supplement is available for figure 2:

**Figure supplement 1.** Sts5 protein contains an intrinsically disordered domain.

developing a rounded cell shape at 36°C (*Figure 4—figure supplement 2A,d*). Consistent with increased Ssp1 protein levels playing a role in promoting abnormal morphogenesis, the aberrant morphological phenotype of sts5 mutants is partially suppressed by loss of *ssp1* (*Matsusaka et al.,*

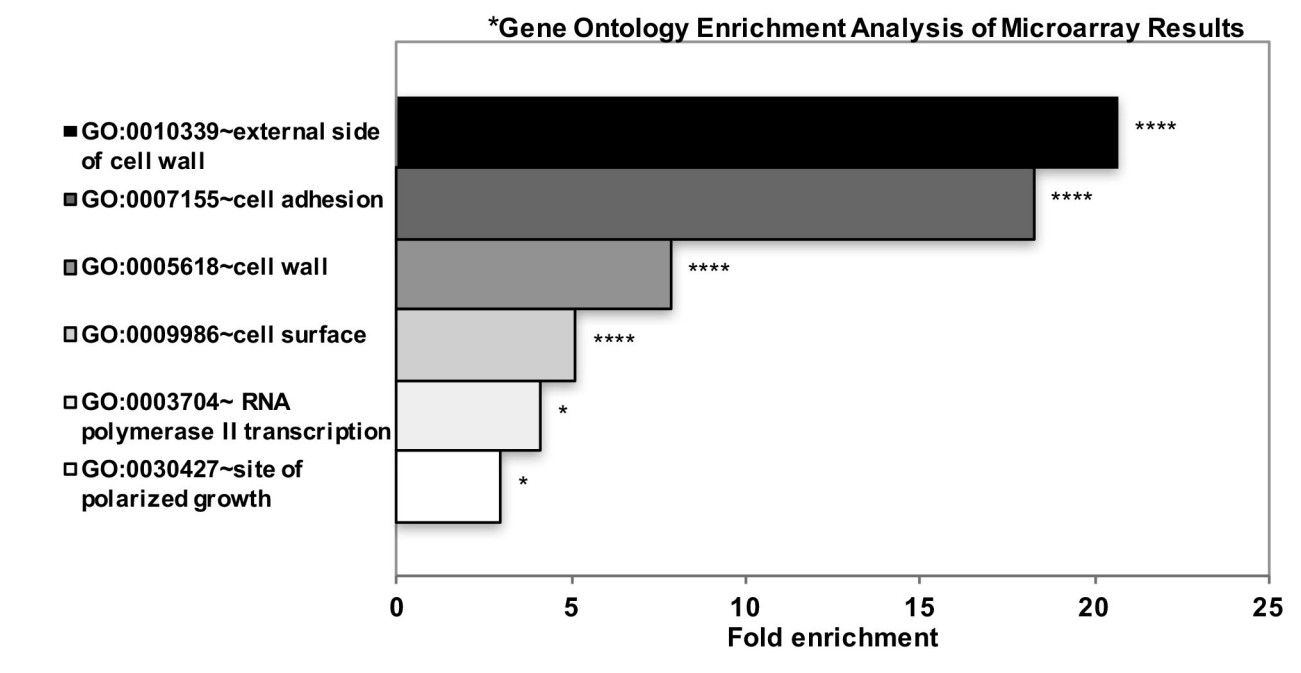

## mRNAs detected at higher levels in *sts5Δ* cells by microarray analysis*
## (The full list is available in Figure 3—Source Data 1.)

| Gene | Ratio sts5Δ/WT | †5' UTR consensus | Function |
|---|---|---|---|
| *cmk2* | 2.26 | yes | CaMK, polarized growth and NETO control |
| *lkh1* | 2.15 | yes | LAMMER kinase, polarized growth and NETO control |
| *ssp1* | 2.13 | yes | CaMKK, polarized growth and NETO control |
| *psu1* | 2.61 | yes | Beta glucosidase, cell wall biogenesis |
| *efc25* | 2.02 | yes | Ras1 GEF, polarity, regulation of Ras1 activity and Cdc42 activation |
| *tea5/ppk2* | 1.95 | yes | pseudokinase, polarized growth and NETO control |
| *ksg1* | 1.94 | yes | PDK1 kinase, cell wall integrity and septation control |

**Figure 3.** mRNAs detected at higher levels in sts5Δ cells by microarray analysis. Total mRNA was extracted from *sts5Δ* and control cells for microarray analysis. A complete list of mRNAs increased in *sts5Δ* cells (≥1.9 fold) is shown in *Figure 3—source data 1*. Several of these mRNAs have established functions in bipolar growth activation and contain putative Sts5-binding sites in their 5' UTRs (*Hogan et al., 2008*; *Wanless et al., 2014*). *Gene ontology enrichment analysis of terms that are significantly enriched among the set of mRNAs with *sts5Δ*/WT ratio ≥1.90 in the microarray results. Fold enrichment plotted per gene ontology category among all significant terms ($P<0.05$, modified Fisher Exact P-value with the Benjamini P-value correction) for Cellular Compartment (CC), Biological Process (BP) and Molecular Function (MF) Gene Ontology terms. †Sts5 binding site: HNNYAHTCHWW (where H = A,T,C / N = A,T,C,G / Y = C,U / W = T,A).

The following source data is available for figure 3:

**Source data 1.** Microarray analysis results.

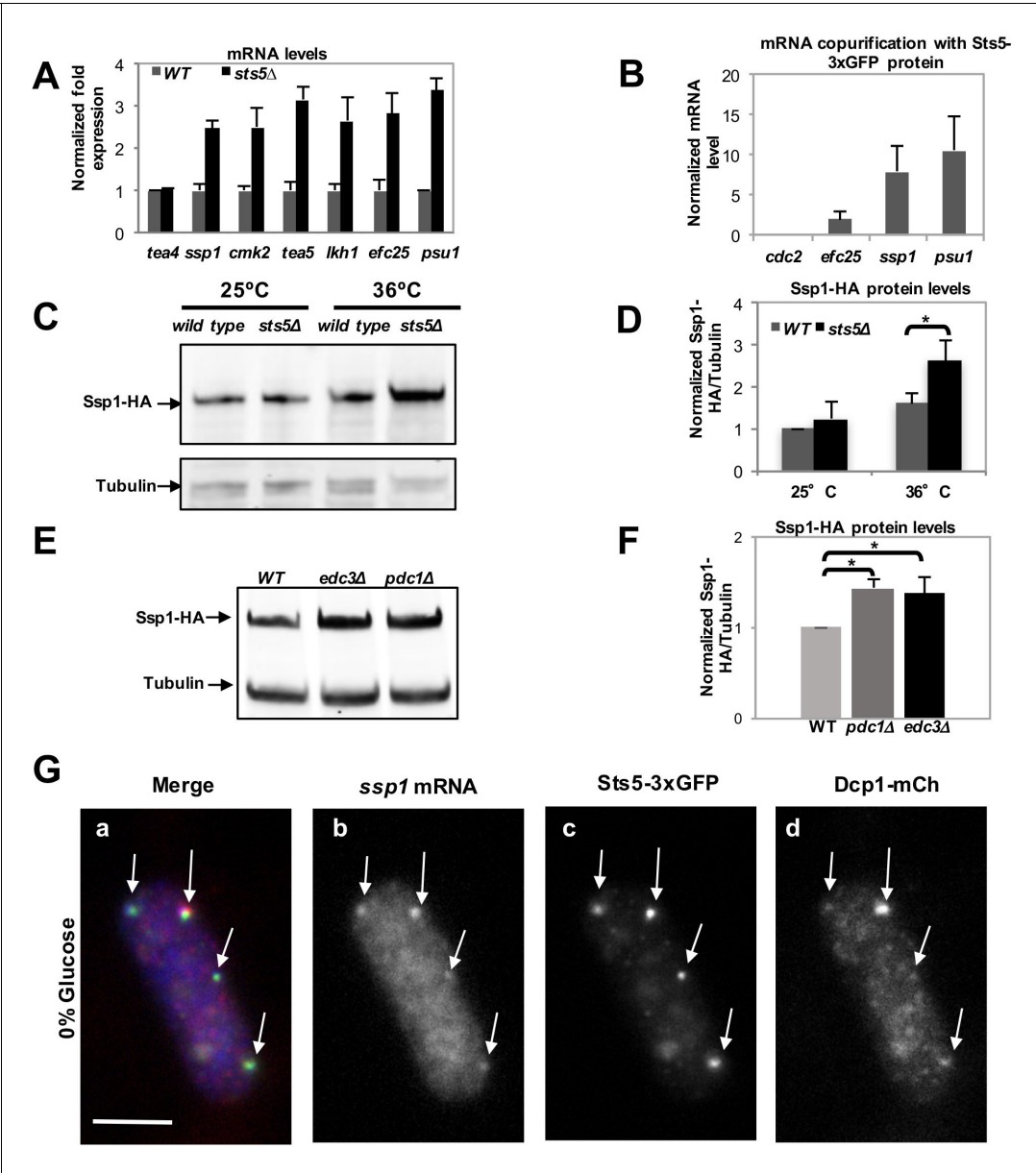

**Figure 4.** Loss of Sts5 leads to increased levels of mRNAs involved in growth control and bipolar growth activation. (A) qPCR analysis confirmation that several of these transcripts are more abundant in the *sts5Δ* strain as compared to control cells based on 3 independent experiments. Tea4 is shown as an example of a transcript that is not altered. Housekeeping genes were *nda3*, *act1*, *cdc2*, and *cdc22*. Error bars indicate SD. (B) Interaction of *ssp1*, *efc25*, and *psu1* mRNAs with Sts5-3xGFP as established by co-immunoprecipitation with Sts5-3xGFP followed by qPCR as described in the Materials and Methods. *cdc2* is shown as an example of a transcript that does not interact with Sts5-3xGFP. Error bars indicate SD. Three independent experiments were performed. (C) Western blotting against Ssp1-HA performed as described in Materials and Methods in WT and *sts5Δ* cells cultured in YE medium at 25°C and 36°C. Tubulin levels were determined as a loading control. (D) Quantification of Ssp1-HA/Tubulin ratio normalized to WT levels was based on 3 independent experiments. Change in Ssp1-HA level is significantly greater in *sts5Δ* cells as compared to controls at 36°C (*P* = 0.034, Student's t-test). Error bar=SD. (E) Western blotting against Ssp1-HA performed as described in Materials and Methods in WT and *edc3Δ* and *pdc1Δ* cells cultured in supplemented minimal medium at 30°C. Tubulin levels were determined as a loading control. (F) Quantification of Ssp1-HA/Tubulin ratio normalized to WT levels at 25°C based on 3 independent experiments. Change in Ssp1-HA level is significantly greater in *edc3Δ* (*P* = 0.018, Student's t-test) and *pdc1Δ* (*P* = 0.0154, Student's t-test) cells as compared to controls. Error bar = SD. (G) RNA FISH visualization of *ssp1* mRNA in fixed cells cultured for 20 min in supplemented minimal medium containing 0% glucose. Hybridization used 20-mer DNA oligos (Stellaris) labeled with Quasar 705 fluorochromes. Bar = 5 μm.

The following figure supplements are available for figure 4:

**Figure supplement 1.** Overexpression of Psu1 suppresses the temperature-sensitive growth defect of *orb6-25* mutant cells.

*Figure 4 continued on next page*

*Figure 4 continued*

**Figure supplement 2.** Deletion of *sts5* alters cell shape and Myc-Efc25 protein levels.

**Figure supplement 3.** Extent of colocalization between *ssp1* mRNA, Sts5-3xGFP, and Dcp1-mCherry in fixed cells cultured in the presence and absence of glucose.

*1995*; *Toda et al., 1996*). In addition, we determined that loss of *sts5* leads also to increased levels of Myc-Efc25 proteins as compared to the control *sts5+* cells (*Figure 4—figure supplement 2B,C*). Taken together, these findings indicate a role for Sts5 in regulating the cellular abundance of specific mRNAs, affecting cell morphology in particular during cell stress.

Next, we tested if loss of P-body components affects the protein levels of Ssp1. We assayed the levels of Ssp1 protein in *pdc1Δ* and *edc3Δ* mutant cells. Both Pdc1 and Edc3 are P-body components and bind to the mRNA decapping complex catalytic subunit Dcp2 (*Fromm et al., 2012*; *Wang et al., 2013*). *pdc1* encodes an mRNA decapping scaffolding protein and *pdc1Δ* mutants display reduced levels of P bodies and reduced mRNA decapping (*Wang et al., 2013*). *edc3* encodes an enhancer of mRNA decapping and *edc3Δ* mutants display decreased decapping of nuclear-transcribed mRNA (*Fromm et al., 2012*; *Wang et al., 2013*). We found that levels of Ssp1 protein are increased, as compared to tubulin control, in both the *pdc1Δ* and *edc3Δ* mutant backgrounds (*Figure 4E,F*). This effect is seen even in the absence of starvation, during exponential cell growth, consistent with the idea that P-body components modulate mRNA abundance even in the absence of large P-body formation (*Decker et al., 2007*; *Eulalio et al., 2007*).

Finally, we tested if *ssp1* mRNA localizes to the P-bodies during glucose starvation, using RNA FISH methodology. We found that *ssp1* mRNA readily co-localizes to Sts5- and Dcp1-containing granules in cells re-diluted in minimal medium (EMM) lacking glucose for 20 min (*Figure 4G*; *Figure 4—figure supplement 3A–C*). Conversely, no co-localization was observed in control cells re-diluted in growth medium containing 2% glucose (*Figure 4—figure supplement 3A-C*). Our data reveal a role for Sts5 and P-body components in regulating the levels of *ssp1* mRNA and Ssp1 protein.

## Orb6 kinase activity inhibits Sts5 recruitment and localization to P-bodies

To test the role of Orb6 kinase in the control of Sts5, we used the analog-sensitive *orb6-as2* mutant to determine whether loss of Orb6 kinase function alters the localization of Sts5. In *orb6-as2* cells treated with the 1-NA-PP1 inhibitor, Sts5-3xGFP rapidly coalesces into cytoplasmic puncta that colocalize with the P-body marker Dcp1-mCherry (*Figure 5A, d, h, and l*; see quantification in *Figure 5B*), while DMSO-treated cells exhibit diffuse cytoplasmic localization of Sts5-3xGFP and Dcp1-mCherry (*Figure 5A, c, g, and k* and *Figure 5B*), supporting the idea that Orb6 kinase negatively regulates Sts5-3xGFP recruitment into RNP granules and Sts5 co-localization with P-bodies.

Recent work has shown that the formation of RNP granules often occurs as a result of liquid-liquid phase transition controlled by the concentration of RNP component proteins (*Kroschwald et al., 2015*; *Lin et al., 2015*; *Elbaum-Garfinkle et al., 2015*; *Patel et al., 2015*; *Kato et al., 2012*; *Hyman et al., 2014*; *Brangwynne et al., 2009*; *Lee et al., 2013*; *Brangwynne, 2013*; *Becker and Gitler, 2015*). To establish if Sts5 has a role in promoting P-body formation, we inhibited Orb6-as2 kinase and measured the number of Dcp1-mCherry containing granules in the presence or absence of Sts5. Interestingly, we found that the number of Dcp1-mCherry granules was significantly reduced in the *sts5Δ* background, indicating that Sts5 recruitment has a role in promoting P-body formation (*Figure 6A–C*). We also observed residual formation of Dcp1-mCherry dots, suggesting that Orb6 kinase can induce P-body formation via Sts5-dependent as well as Sts5-independent mechanisms. Conversely, Dcp1-mCherry granules were not induced by DMSO or 1-NA-PP1 in *orb6+* control and *sts5Δ* cells (*Figure 6—figure supplement 1A*).

To test the role of Orb6 kinase in preventing the degradation of Sts5-regulated mRNAs, qPCR analysis was performed to probe the levels of specific transcripts following Orb6-as2 kinase

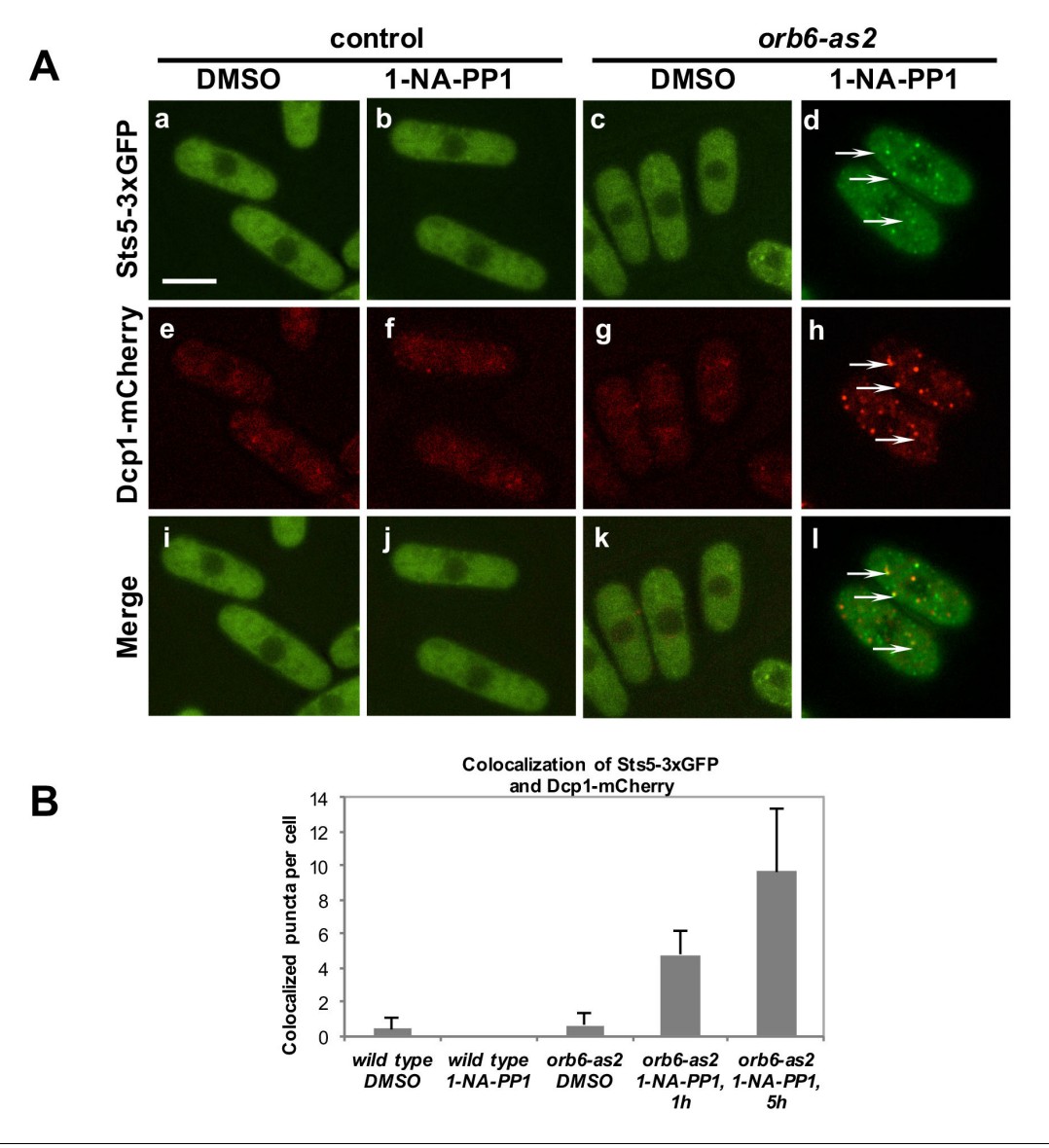

**Figure 5.** Orb6 kinase inhibits Sts5 recruitment and localization to P-bodies. (**A**) Loss of Orb6-as2 kinase activity leads to Sts5-3xGFP recruitment into puncta that colocalize with the P-body marker Dcp1-mCherry. Cells were treated with inhibitor or DMSO for 1 hr (shown) and 5 hr. Bar = 5 μm. (**B**) Quantification of three sets of experiments as shown in **A**.

inhibition with 1-NA-PP1. We found that mRNA levels of *ssp1, efc25* and *psu1* declined upon Orb6 kinase inhibition (*Figure 6D*). Consistent with the idea that Orb6 kinase prevents degradation and translational repression of Sts5-regulated mRNAs, immunoblotting analysis showed that Ssp1-HA protein levels decrease in *orb6-as2* cells upon inhibition with 1-NA-PP1 (*Figure 6E,F*). Normal Ssp1-HA protein levels were maintained in *sts5Δ orb6-as2* cells upon inhibition of Orb6-as2 kinase, in accordance with the findings that *sts5* suppresses the viability phenotype of *orb6* mutants and *sts5Δ* cells accumulate *ssp1* mRNA. These observations held true also for Ssp1-HA protein levels in temperature-sensitive *orb6-25* cells cultured at the non-permissive temperature (36°C) in the presence and absence of Sts5 (*Figure 6—figure supplement 1B–D*). Similarly to Ssp1-HA, Myc-Efc25 protein levels also declined in *orb6-as2* mutants grown in the presence of 1-NA-PP1 (*Figure 6—figure*

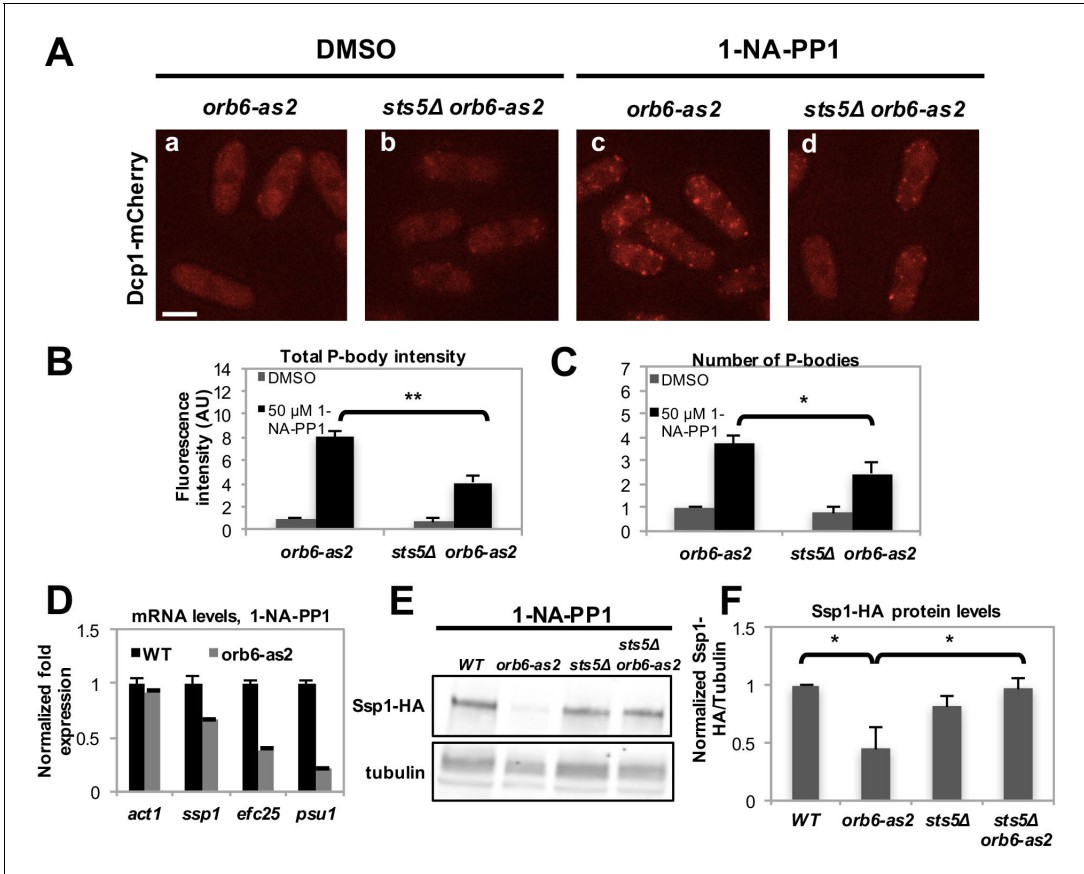

**Figure 6.** Orb6 kinase inhibits Sts5-dependent P-body formation and translational repression. (A) Dcp1-mCherry localization in *orb6-as2* (a, c) compared with *sts5Δ orb6-as2* (b, d) cells grown in supplemented minimal medium in the presence of 50 μM 1-NA-PP1 (c and d) or DMSO (a and b) for 1 hr. Loss of Sts5 in the *sts5Δ orb6-as2* strain decreases the number of P-bodies induced by Orb6 kinase inhibition. Images are deconvolved projections from 12 Z-stacks separated by a step size of 0.3 μm. Bar = 5 μm. (B) Quantification of the experiment shown in A based on 3 independent experiments (n > 24 cells per sample in each experiment). The number of P-bodies per cell was significantly lower in *sts5Δ orb6-as2* cells as compared to *orb6-as2* cells upon Orb6 kinase inhibition relative to DMSO-treated *orb6-as2* cells (P = 0.0186, Student's t-test). No significance difference was observed when comparing *orb6-as2* vs *sts5Δ orb6-as2* cells treated with DMSO, P = 0.2458 (Student's t-test). Error bars indicate SD. (C) Quantification of the experiment shown in A based on 3 independent experiments (n > 24 cells per sample in each experiment). The total P-body fluorescence intensity per cell was significantly lower in *sts5Δ orb6-as2* cells as compared to *orb6-as2* cells upon Orb6 kinase inhibition relative to DMSO-treated *orb6-as2* cells (P = 0.0013, Student's t-test). No significance difference was observed when comparing *orb6-as2* vs *sts5Δ orb6-as2* cells treated with DMSO (P = 0.1837, Student's t-test). Error bars indicate SD. (D) mRNA levels of Sts5-regulated transcripts decrease upon Orb6-as2 kinase inhibition as compared to control, as established by qPCR analysis based on 3 independent experiments. *act1* is shown as an example of a transcript that is not altered. Housekeeping genes were *nda3, cdc2*, and *cdc22*. Error bars indicate SD. (E) Ssp1-HA protein levels in control, *orb6-as2, sts5Δ*, and *sts5Δ orb6-as2* cells cultured in the presence of 50 μM 1-NA-PP1 inhibitor in supplemented minimal medium at 25°C. Tubulin levels were determined as a loading control. (F) Quantification of Ssp1-HA/Tubulin in 4 independent experiments, as shown in E, normalized to wild-type levels. Ssp1-HA levels are significantly reduced upon Orb6-as2 kinase inhibition (P = 0.031), and are restored to wild-type levels in the *sts5Δ orb6-as2* strain (P = 0.023). P values were determined using analysis of variance (ANOVA) with SPSS statistics package 22.0, followed by Games-Howell test. Error bars indicate SD.

The following figure supplements are available for figure 6:

**Figure supplement 1.** Orb6 kinase inhibits Sts5-dependent translational repression.

**Figure supplement 2.** Overexpression of Orb6 inhibits Sts5 granule assembly, and Sts5 plays a role in P-body formation.

*supplement 1E*), and *sts5Δ* abolished the reduction of Efc25 levels in *orb6-as2* mutants (*Figure 6—figure supplement 1E*).

Finally, we tested if Orb6 kinase over-expression alters Sts5 recruitment and P-body formation following glucose deprivation. Indeed, we found that cells over-expressing Orb6 kinase display

significantly smaller Sts5- and Dcp1-containing particles than control cells, following growth for 1 hr in minimal medium without glucose (*Figure 6—figure supplement 2A*). Further, consistent with Sts5 having a role in promoting, at least in part, P-body formation during nutritional stress we found that *sts5Δ* cells form significantly smaller and dimmer Dcp1-containing P bodies, as compared to control cells, following 1 hr growth in glucose deprivation conditions (shift from 2% to 0.01% glucose) (*Figure 6—figure supplement 2B–D*).

Together, these findings support the idea that Orb6 kinase prevents Sts5 recruitment to Dcp1-containing granules and attenuates P-body formation, in a manner that is at least in part Sts5-dependent. Additionally, the function of Orb6 kinase activity has the effect of decreasing the degradation of specific mRNAs.

## 14-3-3 protein Rad24 negatively regulates Sts5 recruitment into cytoplasmic puncta

We performed an in-vitro kinase assay by purification of Orb6 kinase regulatory subunit Mob2, as previously reported (*Wiley et al., 2003*; *Das et al., 2009, 2015*), using bacterially expressed Sts5. We found that the immunoprecipitate readily phosphorylates Sts5 (*Figure 7A*), suggesting that Orb6 kinase phosphorylates Sts5. Sts5 contains several putative NDR kinase consensus sequences (*Hao et al., 2008*; *Mazanka et al., 2008*; *Gógl et al., 2015*) that are consistent with 14-3-3 binding sites (RxxS) when phosphorylated (*Yaffe et al., 1997*). We previously showed that 14-3-3 protein Rad24 has a role in negatively regulating another Orb6 substrate, Cdc42 GEF Gef1 (*Das et al., 2015*). In order to establish whether Sts5 may be subject to regulation by Rad24, we performed a pull-down assay to test whether Sts5 binds Rad24. This assay confirmed that Sts5-HA physically associates with Rad24-GST and not with GST alone (*Figure 7B*). Consistent with Rad24 negatively regulating Sts5 recruitment, we found that Sts5-3xGFP forms cytoplasmic puncta in *rad24Δ* mutants even when cultured in rich medium (YE) in the presence of glucose (*Figure 7C,D*). This effect occurs in growth conditions where cells are not starved and P-body formation is not strongly induced in either *rad24Δ* or control cells (*Figure 7C*). Accordingly, we found that *ssp1* mRNA levels do not significantly change in *rad24Δ* mutants as compared to control cells (*Figure 7E*).

Finally, we tested the effects of Orb6 kinase activity inhibition on the association of Rad24 to Sts5. We found that that Sts5-3xGFP association with GST-Rad24 is abrogated by inhibition of Orb6-as2 kinase activity following exposure of *orb6-as2* cells to 1-NA-PP1, and not in *orb6-as2* cells exposed to DMSO or in control *orb6+* cells (*Figure 7F*; see quantification in *Figure 7—figure supplement 1*). Collectively, our findings indicate that Sts5 protein associates with 14-3-3 protein Rad24 in a manner that is dependent on Orb6 kinase activity, and that this association prevents Sts5 coalescence into cytoplasmic puncta.

## Active Orb6 kinase localization spatially anti-correlates with Sts5 recruitment into puncta in interphase cells

Orb6 kinase localization is enriched at the growing cell tips during interphase, in a manner that depends on the pattern of growth of the cell (*Figure 8A*; *Figure 8—figure supplement 1A,a,b,c and B*) (*Verde et al., 1998*). In smaller cells, which grow in a monopolar manner (M), Orb6 kinase localization is higher at the old growing end and lower at the non-growing new end (*Figure 8A*, *Figure 8—figure supplement 1A,a*). To establish whether Orb6 kinase has a role in spatially controlling Sts5 in interphase cells during exponential cell growth, we tested the extent of Sts5 recruitment into small granules in smaller, monopolar cells (9.1 μm on average), which grow from the old end only. As shown earlier (See *Figure 2A*), during growth in rich medium, Sts5 localization is generally diffuse. However, a closer inspection indicated that most cells contain a few small Sts5 puncta (*Figure 8B*). We consistently found an increase in the number and intensity of Sts5 puncta at the non-growing end of smaller cells (*Figure 8C*; *Figure 8—figure supplement 1A,d,e and f*; *Figure 8—figure supplement 1B*), indicating an inverse correlation between Orb6 kinase localization at the growing tip and Sts5 aggregation (*Figure 8A–C*).

To further investigate this effect, we visualized Sts5-3xGFP in longer *tea1Δ* cells that grow from one end only (*Snell and Nurse, 1994*; *Verde et al., 1995*). *tea1Δ* cells display monopolar Orb6-GFP localization at the only growing cell tip (*Figure 8D*; *Figure 8—figure supplement 2A,a*). In these cells, Sts5-3xGFP recruitment into small puncta was clearly seen increasing towards the non-growing

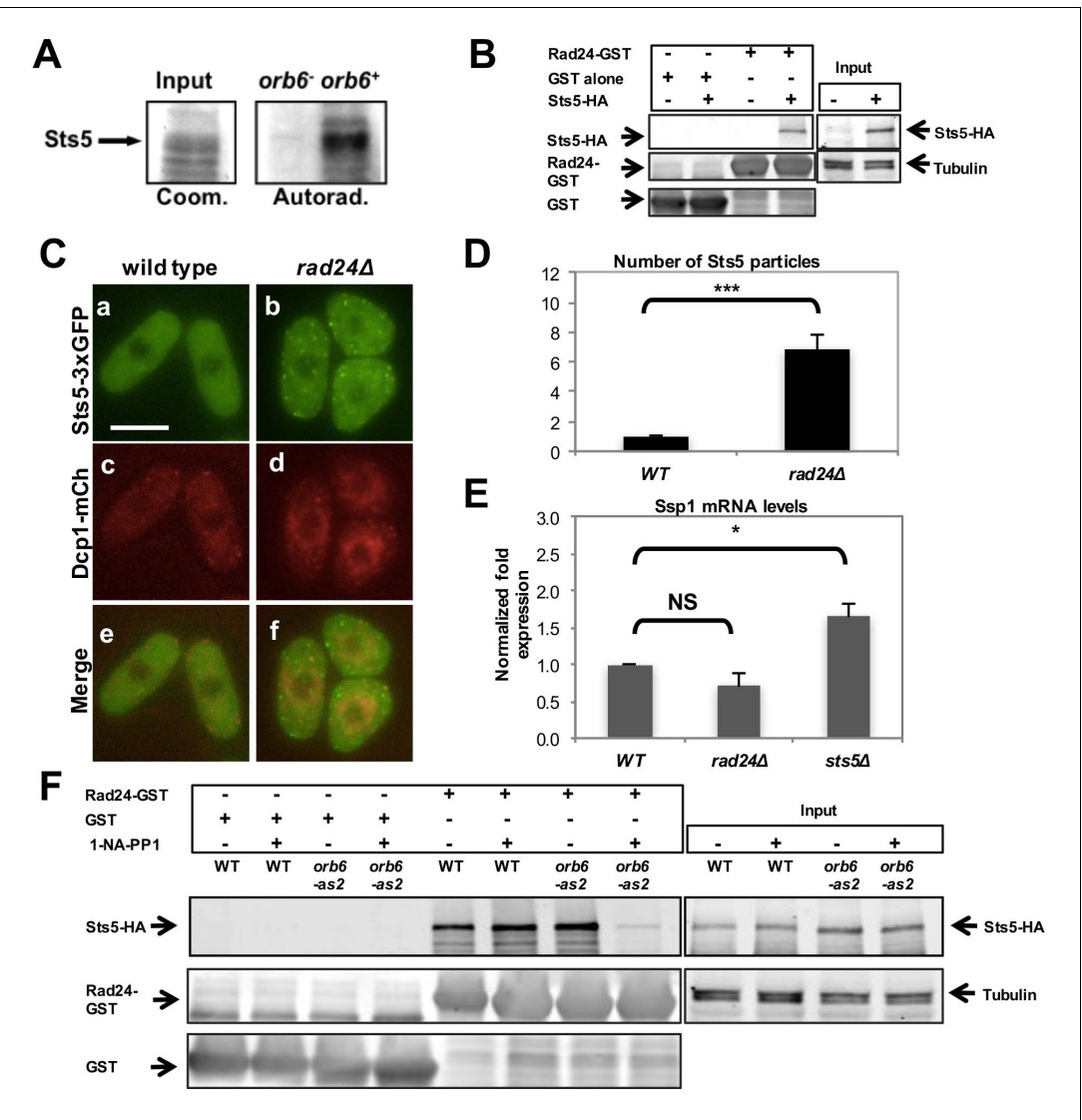

**Figure 7.** 14-3-3 protein Rad24 negatively regulates Sts5 recruitment into puncta. (**A**) Orb6 kinase phosphorylates Sts5 *in vitro*. Mob2-associated Orb6 kinase was immunoprecipitated for a kinase assay as described in the Materials and Methods and incubated with bacterially expressed Sts5 in the presence of $[\gamma^{32}P]$ATP. (**B**) Endogenously expressed Sts5-HA co-purifies with bacterially expressed GST-Rad24 but not with GST alone in a pull-down assay. Three independent experiments were performed. (**C**) Sts5-3xGFP and Dcp1-mCherry aggregation in 2% glucose YE in WT vs *rad24Δ* cells. Images are deconvolved projections from 12 Z-stacks separated by a step size of 0.3 μm. Bar = 5 μm. (**D**) Quantification of the experiment shown in **C** based on 3 independent experiments (n > 27 cells per strain in each experiment). The number of Sts5 particles is significantly higher in *rad24Δ* relative to wild-type control cells (*P* = 0.0005, Student's t-test). Error bars indicate SD. (**E**) qPCR analysis showing *ssp1* mRNA levels are unchanged in *rad24Δ* cells compared with WT (*P* = 0.160) and increased in *sts5Δ* cells compared with WT (*P* = 0.044). When comparing *sts5Δ* with *rad24Δ* cells, P=0.006. P values were determined using analysis of variance (ANOVA) with SPSS statistics package 22.0, followed by Games-Howell post-hoc test. Housekeeping genes were *nda3, act1*, and *cdc2*. Error bars indicate SD. Three independent experiments were performed. (**F**) Physical association between endogenously expressed Sts5-HA and bacterially expressed GST-Rad24 is lower upon inhibition of Orb6-as2 with 50 μM 1-NA-PP1 compared with DMSO treatment (lanes 7 and 8). Sts5-HA association with GST-Rad24 remains unchanged in wild-type cells in the presence or absence of the inhibitor (lanes 5 and 6). GST-only control is shown in lanes 1–4.

The following figure supplement is available for figure 7:

*Figure 7 continued on next page*

*Figure 7 continued*

**Figure supplement 1.** Quantification of the physical association between endogenously expressed Sts5-HA and bacterially expressed GST-Rad24.

cell tip (*Figure 8E-F*; *Figure 8—figure supplement 2A,b–f*; *Figure 8—figure supplement 2B*: asterisk marks the growing tip).

Collectively, our findings indicate that Orb6 kinase activity negatively regulates Sts5 recruitment into cytoplasmic puncta, via interaction with 14-3-3 protein Rad24, in a manner that is spatially significant: an asymmetry of Orb6 distribution between growing and non-growing tips correlates with an asymmetry in Sts5 recruitment.

## Role of Orb6 kinase in the control of Sts5 during cell separation

Orb6 kinase activity is repressed during mitosis by the septation-initiation network (SIN), which triggers cytokinesis (*Kanai et al., 2005*; *Gupta et al., 2013*, *Gupta et al., 2014*). SIN signaling remains active until completion of cytokinesis, which is marked by closure of the contractile ring and a fully formed cell septum (*García-Cortés and McCollum, 2009*; *Alcaide-Gavilán et al., 2014*). After cytokinesis, the primary septum must be degraded for cell separation to occur. The derepression of MOR signaling, and of Orb6 activity, that results from inactivation of the SIN pathway promotes cell separation upon completion of cytokinesis (*Gupta et al., 2014*). Consistent with Orb6 kinase re-activation, Sts5-3xGFP puncta dissipate when the septum is fully closed and the actomyosin contractile ring protein Rlc1, encoding myosin II regulatory light chain (*Le Goff et al., 2000*; *Wu et al., 2006*), disappears from the plane of cell division (*Figure 8G,H*; *Wei et al., 2016*). Orb6-GFP is still physically present at the site of cell division during septum formation and cytokinetic ring constriction (*Figure 8—figure supplement 1A,c*) while it is enzymatically repressed by SIN signaling (*Kanai et al., 2005*; *Gupta et al., 2013*, *Gupta et al., 2014*; *García-Cortés and McCollum, 2009*; *Alcaide-Gavilán et al., 2014*). Consistent with a role for Orb6 kinase reactivation in mediating Sts5-3xGFP granules dissipation, Orb6 kinase inhibition maintains Sts5-3xGFP granules even following actin ring closure and Rlc1 disappearance (*Figure 8—figure supplement 3A,B*).

Supporting a role for Orb6 kinase and its substrate target Sts5 in cell separation, microarray analysis found that Sts5 negatively regulates transcripts that encode cell wall proteins with potential functions in cell separation. Transcripts for a predicted β-1,3 glucanase (encoded by SPBP23A10.11C) and predicted β-glucosidase Psu2 are more abundant in *sts5Δ* cells (See *Figure 3—source data 1*), consistent with a role for β-1,3 glucan degradation in the primary septum during the process of cell separation (*Martín-Cuadrado et al., 2003*). In addition, *sts5Δ* cells accumulate transcripts of the transcription factor Mbx1 (See *Figure 3—source data 1*) that cooperates with the transcription factor Ace2 to promote expression of endo-glucanase Agn1, a hydrolytic enzyme involved in septum degradation (*Suárez et al., 2015*).

Thus, it is possible that Sts5 recruitment into puncta during mitosis, mediated by SIN pathway-dependent inhibition of Orb6 kinase, functions to translationally repress mRNAs encoding cell wall hydrolytic enzymes that would interfere with the deposition of the primary septum. Consistent with this idea, we found that *sts5Δ* cells are prone to rupture at the site of cell separation (*Figure 8—figure supplement 4A,d and B*) similarly to cells that express ectopically active Orb6 during mitosis (*Gupta et al., 2013*, *Gupta et al., 2014*).

## Sts5 restrains bipolar growth activation during exponential cell proliferation and during nutritional stress

When cultured at 25°C, *sts5Δ* cells appear normal with a cylindrical shape (*Toda et al., 1996*). However, we found that *sts5Δ* cell cultures display an increased percentage of cells growing in a bipolar fashion, as compared to similarly sized control cells, under exponential growth conditions (optical density at 595 nm <0.4) in both rich (*Figure 9A,B*) as well as in minimal medium (*Figure 9—figure supplement 1A*). These findings suggest that Sts5 has a function in partially constraining growth at the new end during the exponential growth phase (OD<0.4), without affecting overall growth rates

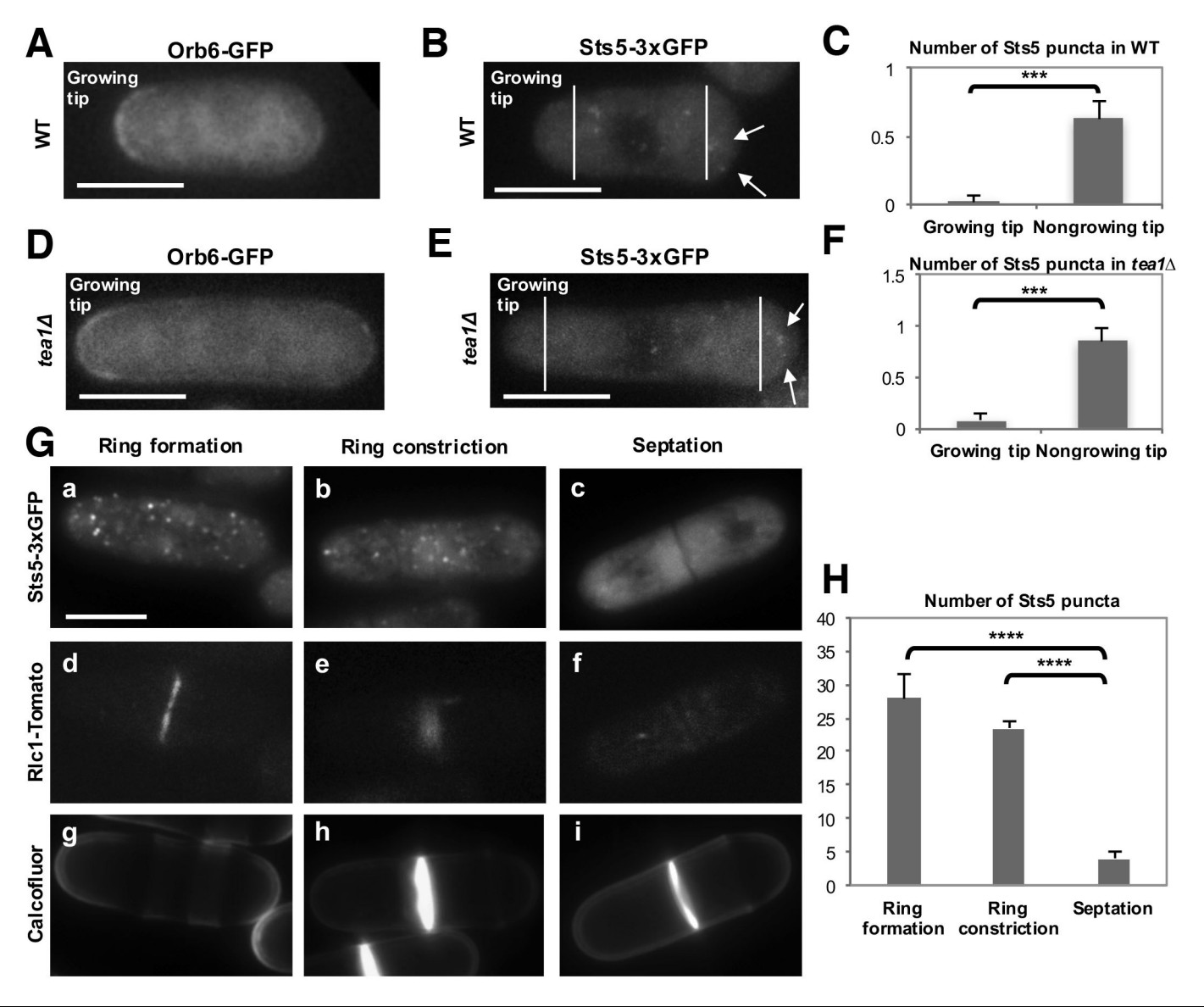

**Figure 8.** Role of Orb6 kinase in Sts5 granule assembly during the cell cycle. (A–F) Active Orb6 kinase localization spatially anti-correlates with Sts5 recruitment into puncta in interphase cells. A. Orb6-GFP localizes to the growing cell tip in small monopolar wild-type cells. Orb6-GFP is enriched at the growing old cell end as compared to the non-growing new cell end. Bar = 5 μm. (B) Sts5-3xGFP aggregation increases towards the new cell end in monopolar wild-type cells. Images are deconvolved projections from 12 Z-stacks separated by a step size of 0.3 μm. Bar = 5 μm. (C) The average number of Sts5-3xGFP puncta per cell at the non-growing new end is significantly higher as compared to the growing old end ($P<0.0001$, Student's t-test). Error bars denote SD. Three independent experiments were performed (N = 31 cells). (D) Orb6-GFP localizes to the growing cell tip in *tea1Δ* cells. Bar = 5 μm. (E) Sts5-3xGFP recruitment onto puncta increases towards the non-growing tip in *tea1Δ* cells. Images are deconvolved projections from 12 Z-stacks separated by a step size of 0.3 μm. Bar = 5 μm. (F) The average number of Sts5-3xGFP puncta per cell at the non-growing end in *tea1Δ* cells is significantly higher as compared to the growing old end ($P<0.0009$, Student's t-test). Error bars denote SD. Three independent experiments were performed (N = 24 cells). We used calcofluor staining to identify growing tips and measured monopolar *tea1Δ* cells that were growing from the previous old end, which facilitated definitive identification of the nongrowing cell end. (G–H) Orb6 kinase activity temporally anti-correlates with Sts5 assembly into puncta during mitosis. (G) (a, b and c) Localization of Sts5-3xGFP in cells undergoing cell division; (d, e and f) visualization of Rlc1-Tomato; (g, h and i) calcofluor staining of cell wall and septum. Bar = 5 μm. (H) Quantification of the number of Sts5 puncta in dividing cells during cytokinetic ring formation, ring constriction, and septation. Ring formation vs septation, $P<0.0001$; ring constriction vs septation, $P<0.0001$; ring formation vs ring constriction $P = 0.588$ (N>20 cells per condition). P values were determined using analysis of variance (ANOVA) with SPSS statistics package 22.0, followed by Tukey's HSD post-hoc test. Three independent experiments were performed.

The following figure supplements are available for figure 8:

*Figure 8 continued on next page*

*Figure 8 continued*

**Figure supplement 1.** Additional images of Orb6-GFP and Sts5-3xGFP localization in monopolar WT cells and quantification of total Sts5-3xGFP granule intensity at growing and nongrowing tips.

**Figure supplement 2.** Additional images of Orb6-GFP and Sts5-3xGFP localization in monopolar *tea1Δ* cells and quantification of total Sts5-3xGFP granule intensity at growing and nongrowing tips.

**Figure supplement 3.** Orb6 kinase inhibition prevents the dissolution of Sts5-3xGFP puncta after completion of mitosis.

**Figure supplement 4.** *sts5Δ* cells display increased cell lysis during cell separation.

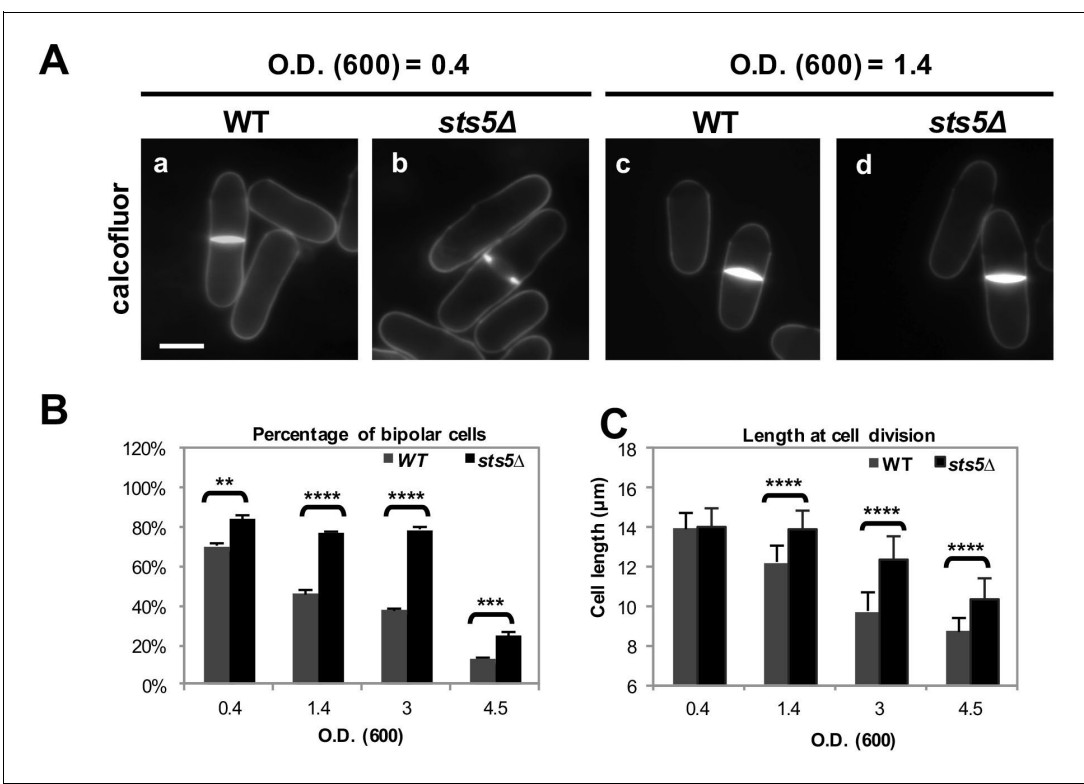

**Figure 9.** Sts5 modulates bipolar growth activation during exponential cell proliferation and during nutritional stress. (**A**) *sts5Δ* cells display a delayed morphological response to nutritional stress induced by high cell density as compared with wild type cells. Cells were stained with calcofluor. Bar = 5 μm. (**B**) Quantification of the percentage of bipolar cells in control versus *sts5Δ* cells in the experiment depicted in **A**. Percentage bipolar cells was significantly higher in *sts5Δ* cells versus control cells during exponential growth ($OD_{600}$ <0.4) ($P$ = 0.0013, Student's $t$ test) and at $OD_{600}$ = 1.4 ($P$<0.0001, Student's t test), $OD_{600}$ = 3 ($P$<0.0001, Student's t test), and $OD_{600}$ = 4.5 ($P$ = 0.0003, Students' t test). Error bars indicate SD. At least 3 independent experiments were performed (N>64 for each strain per cell density condition). Cells undergoing cell division were not included. (**C**) Quantification of cell size (defined as cell length at division) in control versus *sts5Δ* cells in the experiment depicted in **A**. Cell size was significantly longer in *sts5Δ* cells versus control at $OD_{600}$ = 1.4 ($P$<0.0001, Student's t test), $OD_{600}$ = 3 ($P$<0.0001, Student's t test), and $OD_{600}$ = 4.5 ($P$<0.0001, Student's t test). Error bars indicate SD. At least 3 independent experiments were performed (N>16 for each strain per cell density condition).
The following figure supplement is available for figure 9:

**Figure supplement 1.** Increased bipolarity of *sts5Δ* vs wild-type cells is not due to changes in cell size or overall cell growth.

(*Figure 9—figure supplement 1B*) or cell length at division, which are the same as control cells (*Figure 9C*).

Since the pattern of cell growth is altered by nutritional stress, inhibiting bipolar growth activation and increasing the percentage of monopolar cells, (*Su et al., 1996*; *Yanagida, 2009*; *Yanagida et al., 2011*) we hypothesized that Sts5 recruitment into puncta, a form of RNP granules, might have an adaptive role to modulate the morphological response during nutritional starvation. As cell density increases, *S. pombe* cells respond to limiting nutrient availability by entering mitosis at a shorter cell size (*Costello et al., 1986*; *Su et al., 1996*; *Yanagida, 2009*; *Yanagida et al., 2011*). We found that, whereas wild-type cells divide at a shorter length upon starvation induced by high cell concentration, as determined by optical absorbance at 595 nm (*Figure 9*), *sts5Δ* cells maintain a longer length at cell division as cell concentration increases (*Figure 9A,C*). Similarly, a higher proportion of *sts5Δ* mutants continue to activate bipolar growth, as compared to wild-type cells at the same concentration (*Figure 9B*). These observations suggest that Sts5 has a role in partially constraining bipolar cell growth, a function that is important for cellular adaptation to nutrient limitation. Consistent with this idea, we find that *sts5Δ* cells display decreased viability after prolonged starvation (I.N. and F.V., unpublished observation).

Collectively, our results indicate that NDR kinase Orb6 inhibits the recruitment of mRNA-binding protein Sts5 into cytoplasmic puncta by promoting its interaction with 14-3-3 protein Rad24. Further, Orb6 kinase has a role in negatively controlling P-body formation, in a manner that is at least in part Sts5-dependent. This mechanism controls the levels of mRNAs encoding proteins important for polarized cell growth and cell separation. During interphase, Orb6 inhibits Sts5 recruitment in a manner that is biased towards the old end in small cells, thus promoting normal cell morphogenesis and partially constraining extensile growth at the second, newer cell tip. Extensive Sts5 recruitment into smaller puncta during mitosis and into larger RNP granules during nutritional stress may allow proper septum deposition and modulates morphological adaptation to limiting nutrient availability.

## Discussion

In this article, we define a novel mechanism that spatially regulates polarized cell growth and cell morphology in fission yeast during exponential cell proliferation and in response to environmental stressors, such as increased temperature or cell density. Under exponential growth conditions, fission yeast grow in a monopolar fashion during early interphase and activate growth at the new cell tip once a minimal cell length has been achieved. Different control mechanisms cooperate in the activation of the second tip, a process known as NETO (New End Take Off), including the microtubule-dependent Tea1 complex (*Martin and Arkowitz, 2014*; *Sawin and Nurse, 1998*; *Martin et al., 2005*; *Tatebe et al., 2005*), the availability of Cdc42 regulators (*Coll et al., 2003*; *Tatebe et al., 2008*; *Das et al., 2012*), cell transcription (*Vjestica et al., 2013*), and a diverse array of signaling kinases (*Koyano et al., 2010*; *Matsusaka et al., 1995*; *Rupes et al., 1999*; *Koyano et al., 2015*; *Grallert et al., 2013*). We have previously shown that NDR kinase Orb6 promotes cell polarity and regulates bipolar growth by spatially restricting the activation of Cdc42 GTPase, a key morphology control factor (*Das et al., 2009*). We recently showed that Orb6 negatively regulates Cdc42 activation by promoting the association of Cdc42 Guanine Exchange Factor (GEF) Gef1 with 14-3-3 protein Rad24, and thus limiting Gef1 activity at the membrane (*Das et al., 2015*). This function has the effect of spatially regulating Cdc42 activation, thus promoting the emergence of cell polarity.

In this article, we describe a genetically separable role for Orb6 kinase in the control of polarized cell growth and cell separation. We report that Orb6 kinase regulates the association of mRNA-binding protein Sts5, another Orb6 substrate target, with 14-3-3 protein Rad24. This association prevents the recruitment of Sts5 into cytoplasmic puncta (see hypothetical model in *Figure 10A*). Orb6 localization varies during the cell cycle, increasing at the cell tips during interphase and at the cell septum during cell division (*Verde et al., 1998*; *Wiley et al., 2003*). In small cells that have not yet undergone NETO, or in the monopolar *tea1Δ* mutant cells, Orb6 is enriched at the one growing cell tip. Consistent with a role for Orb6 in inhibiting Sts5 assembly into puncta, we have observed that Sts5-3xGFP puncta are spatially anti-correlated with Orb6 kinase activity and are preferentially localized near non-growing cell tips, which are depleted of Orb6 kinase (*Figure 10B*). Because Orb6 kinase is enriched at growing cell tips, this method of Sts5 regulation might promote the availability of Sts5-targeted mRNAs for translation near sites of polarized growth, while partially constraining growth at

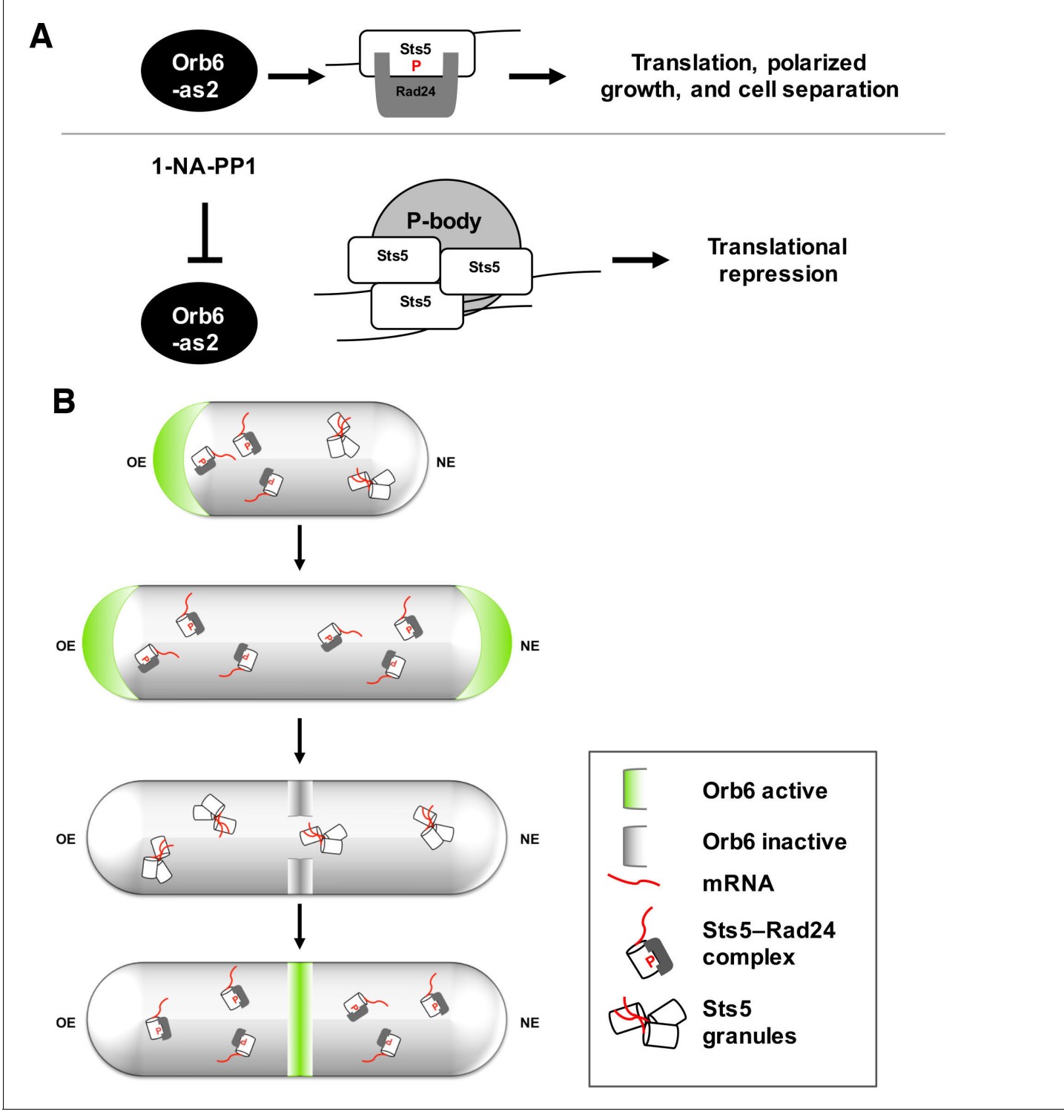

**Figure 10.** A model of spatial control of translational repression and polarized growth by Orb6 kinase and mRNA binding protein Sts5. (A) Orb6 kinase prevents Sts5 recruitment into larger RNP granules by promoting the association between Sts5 and the 14-3-3 protein Rad24. Upon Orb6 kinase inhibition, Sts5 proteins are recruited into larger RNP granules and co-localize with P-bodies, leading to reduced mRNA levels and translational repression. (B) In small monopolar cells Orb6 kinase is localized at the growing old end. Sts5 recruitment in larger granules is observed at the new end, lacking Orb6 kinase activity. In larger bipolar cells Orb6 kinase is localized at both cell tips, and Sts5 recruitment is reduced at both cell ends and throughout the cell. During mitosis, Orb6 kinase is inactivated by the SIN pathway, which allows Sts5 recruitment into larger RNP granules and translational repression. Once cell separation is complete, Orb6 kinase activity resumes, promoting Sts5 disassembly, translational derepression, and cell separation.

the non-growing, newer cell tip. Consistent with this idea, an increased percentage of *sts5Δ* cells exhibit a bipolar pattern of growth, as compared to control cells of similar length.

By microarray and qPCR analysis we found that several transcripts, previously implicated in bipolar growth activation, accumulate in *sts5Δ* cells. These transcripts encode the putative CAMK kinase Cmk2, CAMKK kinase Ssp1, LAMMER kinase Lkh1, pseudokinase Tea5/Ppk2, and PDK1 kinase Ksg1, which promote bipolar growth activation in *S. pombe* (*Koyano et al., 2010*), as well as the Ras1 GEF Efc25, which affects Cdc42 activity (*Papadaki et al., 2002*). In higher eukaryotes, the homologues of these Sts5-regulated transcripts are also implicated in cell growth and morphology. CAMK signaling regulates cytoskeletal organization, plays a role in neuronal development and dendritic spine morphology, and has been shown to be increased in mouse models of cardiac hypertrophy (*Penzes et al., 2008*; *Passier et al., 2000*). The *Drosophila* LAMMER kinase Doa inhibits proliferation of germ cells (*Zhao et al., 2013*). Also, Ras signaling has conserved roles in cytoskeletal organization with implications for cancer development (*Shields et al., 2000*). Activation of PI3K signaling via PDK1 kinase has been implicated in cancer, and PDK1 has been found to regulate cell growth, proliferation, and migration (*Li et al., 2010*; *Mora et al., 2004*).

Consistent with Orb6 kinase inhibiting the extent of Sts5 recruitment during exponential growth, Sts5-containing cytoplasmic puncta assemble during the later stages of mitosis when Orb6 kinase activity is blocked by the SIN pathway (*Figure 10B*) (*Vaggi et al., 2012*; *Gupta et al., 2013*, *Gupta et al., 2014*). Sts5 recruitment during cytokinesis may function to prevent inappropriate translation of proteins involved in septum degradation while the septum is still forming. Indeed, *sts5Δ* cells lyse at the cell septum, in particular upon stress. Once cytokinesis is complete and Orb6 kinase is once again active, Sts5 localization is again diffuse in the cytoplasm, perhaps to allow expression of hydrolases required for cell separation. Consistent with this idea, *orb6* mutants delay cell separation. We did not find obvious induction of P-body formation during mitosis or interphase, likely preventing the degradation of Sts5-regulated transcripts that will eventually be needed for cell separation and polarized cell growth. This result suggests that P-body independent, Sts5-containing RNPs are formed during mitosis and that additional stress signals are required for Sts5 to seed P-body formation under nutritional limitation conditions.

Sts5 bears closest homology to RNA exonucleases such as fission yeast Dis3L2 (*Malecki et al., 2013*). In humans, hDis3L2 is of particular interest because it has been implicated in the congenital Perlman syndrome, which confers fetal overgrowth and susceptibility to Wilms tumor (*Astuti et al., 2012*). Whereas Sts5 lacks crucial amino acids required for exonuclease activity (*Figure 1C*) (*Malecki et al., 2013*; *Uesono et al., 1997*), our work shows that Sts5 promotes mRNA degradation likely by promoting the interaction of Sts5-associated transcripts with P-body components. Similar to Sts5, Dis3L2 also localizes to P-bodies (*Malecki et al., 2013*), as well as the *S. cerevisiae* homologue of Sts5, Ssd1 (*Jansen et al., 2009*; *Kurischko et al., 2011*). These findings suggest that, differently from the related exonuclease Dis3 (*Robinson et al., 2015*), this group of RNA-binding proteins employs a mechanism of mRNA degradation that does not involve interaction with the exosome. However, it is likely that Sts5 and Dis3L2 have different roles in P-body assembly and/or recruitment of mRNAs to P-bodies. Indeed, deletion of the *sts5* homologue *dis3L2* does not suppress the growth defect observed upon Orb6-as2 kinase inhibition (*Figure 1—figure supplement 2B*). In *S. cerevisiae* and *C. albicans*, Sts5 homologue Ssd1, with roles in cell wall hydrolysis (*Jansen et al., 2009*; *Wanless et al., 2014*), fertility (*Bourens et al., 2009*), and transcription (*Lee et al., 2015*), has been proposed to promote localized mRNA translation because of its enrichment at the bud site (*Kurischko et al., 2011*; Lee et al., 2015). We have not observed enrichment of Sts5 at the sites of polarized growth in *S. pombe* cells, which may employ a different strategy for ensuring adequate mRNA localization in daughter cells.

Our work indicates that Sts5 may function as a seed for P-body formation under stress and upon Orb6 kinase inhibition. P-bodies and similar RNPs have been shown to have liquid properties and their formation has been described as condensation by a phase-separation mechanism once P-body components reach critical concentrations (*Kroschwald et al., 2015*; *Lin et al., 2015*; *Elbaum-Garfinkle et al., 2015*; *Hyman et al., 2014*; *Brangwynne et al., 2009*; *Lee et al., 2013*; *Brangwynne, 2013*; *Becker and Gitler, 2015*). It has been proposed that mRNA-binding proteins may play a role in the aggregation of mRNPs into larger structures, especially proteins containing intrinsically disordered domains, which have the potential to promote phase separation by forming multiple weak protein-protein interactions (*Lin et al., 2015*; *Patel et al., 2015*; *Elbaum-*

*Garfinkle et al., 2015*; *Wang et al., 2014*; *Kato et al., 2012*; *Han et al., 2012*; *Malinovska et al., 2013*; *Toretsky and Wright, 2014*; *Kroschwald et al., 2015*). Thus, Sts5 may have the ability to function in mRNP granule formation through interactions with other P-body proteins, and one function of Orb6 phosphorylation may be to limit Sts5 recuitment to prevent inappropriate seeding of P-bodies. Consistent with this idea, Sts5 displays a predicted disordered domain at the N-terminus, which is phosphorylated *in vivo* (*Kettenbach et al., 2015*; *Carpy et al., 2014*; *Koch et al., 2011*; *Wilson-Grady et al., 2008*). Future experiments will address how phosphorylation of this domain modulates the function of Sts5 and affects its properties *in vitro*.

We found that Sts5 has a role in the morphological response to nutritional stress. Wild type *S. pombe* cells mount characteristic morphological responses to changing environmental conditions: increased temperature (*Mitchison and Nurse, 1985*), decreased nutrient availability (*Costello et al., 1986*; *Su et al., 1996*; *Yanagida, 2009*; *Yanagida et al., 2011*), and hyper-osmotic stress decrease the incidence of bipolar growth activation and alter overall cell dimensions (*Rupes et al., 1999*; *Robertson and Hagan, 2008*). The mechanisms that modulate cell morphogenesis and polarized cell growth in response to varying growth and environmental conditions are still poorly understood. We find that *sts5Δ* mutants delay the adaptation to starvation conditions, maintaining bipolar growth and a longer cell length as cell density increases and the medium becomes depleted of nutrients. This effect appears to be further exacerbated at higher temperatures (36°C), where *sts5Δ* cells become enlarged and bloated, with increased protein levels of the CAMKK Ssp1. Our findings indicate that *sts5Δ* mutants have defects in adapting to nutrient deprivation or temperature increase, and fail to manifest the appropriate morphological response to varying extracellular conditions. Thus Sts5 may integrate diverse nutritional and environmental signals to coordinate changes to the pattern of cell growth.

In summary, our results support a role for NDR kinases in the spatial control of polarized cell growth, during cell proliferation and in response to the nutritional environment by mediating the translational availability of specific mRNAs. Future research will seek to identify nutrient-sensitive signaling pathways upstream of Orb6 and define the specific roles of Orb6 kinase and Sts5 in the control of P-body assembly. Due to the conservation of these factors, this work has the potential to open new avenues of research linking nutrient-sensitive signaling and P-body regulation with implications for studies of cancer and neurodegenerative diseases.

## Materials and methods

### Strains and cell culture

*S. pombe* strains used in this study are listed in the supplement in *Supplementary file 1*. All strains used in this study are isogenic to the original strain 972. Cells were cultured in yeast extract (YE) medium or minimal medium (EMM) plus required supplements. Prototrophic strain FV2267 was cultured in unsupplemented EMM. For glucose and nitrogen starvation experiments, cells were washed in glucose-free or nitrogen-free EMM before transfer to EMM either lacking or containing 2% glucose or 0.5% nitrogen, respectively. Exponential growth was maintained for at least eight generations before experimental analysis, and genetic manipulations and analysis were carried out using standard techniques (*Moreno et al., 1991*).

### Isolation of the suppressor mutants from the MOR mutants

Approximately $5 \times 10^7$ cells of the *nak1-125* (KP1-6D), *orb6-25* (DH433-12C), or *mor2-276* mutant (DH107-4C) were spread per one YPD plate (1% yeast extract, 2% polypeptone, 2% dextrose, and 2% agar) containing 10 mg/ml Phloxine B (Sigma-Aldrich, P2759) (called YPDP plate), and the plates were incubated at 35.5°C for 4 days. Spontaneously developed Ts+ colonies at 35.5°C were picked up on YPDP plate and incubated at 35.5°C for 3 days. To investigate the cold sensitivity and cell morphology of the mutants, the Ts+ colonies were replica plated on 2 YPDP plates and incubated at 18°C and 35.5°C. In this screening, we selected Ts+ and cold sick (red colony) at 18°C, and isolated 1, 2, or 1 *sts5* mutant alleles from *mor2*, *nak1*, or *orb6* mutants, respectively. Genetic linkage (allelism) between the suppressors and *sts5* was confirmed by tetrad analysis.

## Fluorescence microscopy

Cells expressing fluorescently tagged proteins were photographed using an Olympus fluorescence BX61 microscope (Melville, NY) equipped with Nomarski differential interference contrast (DIC) optics, a 100X objective (NA 1.35), a Roper Cool-SNAP HQ camera (Tucson, AZ), Sutter Lambda 10 + 2 automated excitation and emission filter wheels (Novato, CA) and a 175 W Xenon remote source lamp with liquid light guide. Images were acquired and processed using the Intelligent Imaging Innovations (Denver, CO) SlideBook image analysis software and prepared with Adobe Photoshop CC (San Jose, CA) and ImageJ64 (U. S. National Institutes of Health) (ImageJ, RRID:SCR_003070). For measurements of Sts5-3xGFP and Dcp1-Cherry puncta, we subtracted the contribution of the cytoplasmic background for each cell as previously described (*Das et al., 2012*). This process was performed using an ImageJ plugin that sets a subtraction threshold to 3 standard deviations from cytoplasmic-region mean. Pilot studies were used to obtain means and standard deviations to be used for sample size estimation before determining how many cells to measure in each independent experiment of Sts5-3xGFP or Dcp1-mCh aggregation. The following formulas were used for sample size estimation, assuming an alpha of 0.05, beta of 0.2, and power of 0.8:

$k = (n_2/n_1) = 1$

$n_1 = [(\sigma_1^2 + \sigma_2^2/K)(z_{1-\alpha/2} + z_{1-\beta})^2] / \Delta^2$

$\Delta = |\mu2-\mu1|$ = absolute difference between two means

$\sigma_1, \sigma_2$ = mean variances

$n_1$ = group 1 sample size

$n_2$ = group 2 sample size

$\alpha$ = probability of type I error (set to 0.05)

$\beta$ = probability of type II error (set to 0.2)

$z$ = critical Z value for a given $\alpha$ or $\beta$

## RNA extraction (for qPCR and microarray)

Cells were grown under normal conditions (eight generations of exponential growth) prior to the start of the experiment. Cells were then treated in accordance with the particular experiment. The RNA was extracted from the yeast using the ZR Fungal/Bacterial RNA MiniPrep kit (Zymo Research). After elution of the RNA, the remaining genomic DNA was digested with TURBO DNA-free (Ambion). The digestion of genomic DNA was confirmed by PCR amplification of the housekeeper genes.

## qPCR analysis

RNA was quantified via NanoDrop, and cDNA was prepared using the iScript cDNA Synthesis Kit (Bio-Rad). The qPCR reaction was done with SsoFast Evagreen Supermix (Bio-Rad) using primers design with Beacon in a Bio-Rad CFX96 Real-Time PCR system. Data was analyzed with Bio-Rad CFX Manager 2.0 software using a regression Cq determination mode. Our housekeeper genes were *nda3*, *act1*, *cdc2*, and *cdc22* (depending on the experiment). Each condition was run at least in triplicate and 3 independent experiments were performed.

## Microarray analysis

RNA was provided to the Oncogenomics Facility (http://sylvester.org/research/shared-resources/laboratory-resources/oncogenomics-core-facility) for the Bioanalyzer to assess RNA quality and amount, followed by microarray hybridization and scanning using the Affymetrix GeneChip Yeast Genome 2.0 Array. Data was then analyzed with MEV (http://www.tm4.org/mev) (TM4 Microarray Software Suite: TIGR MultiExperiment Viewer, RRID:SCR_001915) after conversion to RMA via RMAExpress (http://rmaexpress.bmbolstad.com/) (RMA Express, RRID:SCR_008549).

## Gene ontology enrichment analysis

Gene ontology enrichment analysis was performed using Database for Annotation, Visualization, and Integrated Discovery (DAVID) Bioinformatics Resource 6.7 (DAVID, RRID:SCR_001881).

## Immunoprecipitation of Sts5-3xGFP and identification of associated mRNAs

Cultures were grown of wild-type cells and *sts5-3xGFP* cells for harvesting. Cell pellets were broken in breaking buffer (20 mM Tris-HCl (pH 8.0), 140 mM KCl, 1.8 mM MgCl$_2$, 0.1% NP-40, 0.2 mg/ ml heparin, 0.5 mM DTT, protease inhibitors (complete EDTA-free protease inhibitor cocktail tablets (Roche Applied Science)), 100 U/ml Rnasin Plus (Promega)) with a Savant FastPrep FP120 bead beater. The Sts5 protein was then immunoprecipitated with anti-GFP (Roche; RRID:AB_390913) and protein G magnetic resin (Invitrogen). After extensive washing of the resin with wash buffer (20 mM Tris-HCl (pH 8.0), 140 mM KCl, 1.8 mM MgCl$_2$, 10% glycerol, 0.5 mM DTT, 0.01% NP-40, 10 U/ml Rnasin Plus, and protease inhibitors in the beginning washes), the RNA was eluted from the resin by treating the resin with proteinase K. The RNA was then purified with a spin column kit (ZR Fungal / Bacteria RNA MiniPrep Kit, Zymo Reseach). After elution of the RNA, the remaining genomic DNA was digested with TURBO DNAse (Ambion) and the digestion was confirmed by PCR. qPCR was then used to determine the relative levels of target mRNA in WT (null IP) versus the Sts5-3xGFP IP.

## Bacterially expressed Sts5 protein purification

Sts5 ORF (a.a. 1–1066) was tagged with N-terminal His6 by cloning into pET15b expression vector. The construct was transformed in BL21 cells, and His6-Sts5 expression was induced by incubation with 1mM IPTG for 1 hr. Native His6-Sts5 was purified using Ni-NTA spin columns (Qiagen) following the manufacturers instructions. Western blot using anti-His6 antibody (Covance; AB_10063707) was performed to confirm the purification of His6-Sts5.

## Mob2-associated kinase assay

*In vitro* kinase assay for phosphorylation of Sts5 was performed as described in *Wiley et al. (2003)*. Briefly, Myc-tagged Mob2 and untagged Mob2 were expressed in *S. pombe* cells grown to mid-log phase at 32°C. Cells lysis was performed using Savant FastPrep FP120 bead beater in HB buffer (25 mm MOPS, pH 7.2, 60 mM β-glycerophosphate, 15 mM p-nitrophenyl phosphate, 15 mM MgCl$_2$ 15 mM EGTA, 1 mM dithiothreitol, 0.1 M sodium vanadate, 1% Triton X-100, 1 mM phenylmethylsulfonyl fluoride, and protease inhibitors (complete EDTA-free protease inhibitor cocktail tablets (Roche Applied Science))). Extracts from cells expressing Myc-tagged Mob2 and from wild-type cells were incubated with Protein A agarose (Sigma-Aldrich) beads bound to rabbit anti-Myc antibodies (Santa Cruz Biotechnology; RRID:AB_631274) for 1 hr, washed twice with HB buffer, and then washed once with kinase buffer (50 mM Tris-HCl, pH 7.5, 100 mM NaCl, 10 mM MgCl$_2$,1 mM MnCl$_2$). The resin was resuspended in 25 µl of kinase buffer containing 10 µCi of [γ-32P]ATP (6000 Ci/mmol) and 20 µM ATP and combined with 5 µl bacterially expressed Sts5. The kinase reaction was stopped after 20 min at 30°C. Proteins were separated on an SDS polyacrylamide gel.

## Western blot analysis of Ssp1-HA levels

The protein extraction was performed as previously described (*Matsuo et al., 2006*). 10-ml cultures of exponentially growing cells were harvested by centrifuging at 5000 rpm for 5 min. The cell pellet was first washed in 1 mL of distilled water and then resuspended in 300 µL of distilled water. Then, 300 µL of 0.6 M NaOH was added, and cells were incubated at room temperature for 10 min and collected by centrifugation. After removing the supernatant, cells were resuspended in modified SDS sample buffer (60 mM Tris HCl pH 6.8, 4% β-mercaptoethanol, 4% SDS, 0.01% bromophenol blue, and 5% glycerol) and boiled for 3 min. The samples were then loaded on 4–15% Mini-PROTEAN TGX gels (Biorad) for routine western analysis.

## Antibodies

The primary antibodies used were mouse monoclonal anti-HA (Covance; RRID:AB_2314672), rabbit polyclonal purified antibody c-Myc (A-14) (Santa Cruz Biotechnology, Inc.; RRID:AB_631274) rat monoclonal anti-α-tubulin [YL1/2] (Novus Biologicals; RRID:AB_305328), mouse monoclonal anti-α-tubulin clone B-5-1-2 (Sigma-Aldrich; AB_477579) and rabbit polyclonal anti-GST (Z-5) (Santa Cruz; AB_631586). The secondary antibodies used were IRDye 800 conjugated anti-mouse antibody (Rockland Immunochemicals, Inc; RRID: RRID:AB_10703265), IRDye 800 conjugated anti-rabbit antibody (Rockland Immunochemicals, Inc; RRID:AB_220152), and IRDye700 conjugated anti-rat antibody

(Rockland Immunochemicals Inc.; RRID: AB_220171). The blots were analyzed using the Odyssey Infrared Imaging system (LI-COR Biosciences).

## Orb6-as2 kinase inhibition

Design and construction of the *orb6-as2* analog-sensitive mutant was previously described (*Das et al., 2009*). Inhibition of Orb6-as2 kinase was carried out using the ATP-analog 1-NA-PP1 (4-Amino-1-tert-butyl-3-(1'-naphtyl) pyrazolo [3,4-d]pyrimidine; Toronto Research Chemicals) diluted in DMSO. In liquid media, a final concentration of 50 µM 1-NA-PP1 was used to achieve Orb6-as2 kinase inhibition. In solid media, the final concentration of 1-NA-PP1 used was 10 µM.

## Rad24 binding assays

Bacterially expressed GST and GST-Rad24 were bound to Glutathione linked sepharose beads or magnetic beads (Pierce). The beads were then mixed with fission yeast protein extract from wild type and Sts5-HA tagged strains incubated for overnight at 4°C. The beads were then washed with TRIS lysis buffer (50 mM TrisCl, PH 7.7; 150 mM NaCl; 5mM EDTA; 5% Glycerol; 1% Triton X; 1 mM PMSF; complete EDTA-free protease inhibitor cocktail tablets (Roche Applied Science)) and separated by SDS polyacrylamide gel and analyzed by western blot using mouse monoclonal Anti-HA antibodies (Covance; RRID:AB_2314672). To inhibit Orb6 kinase, cells were incubated with either DMSO or 50 µM 1-NA-PP1 for 15 min at 32°C.

## RNA fluorescence in situ hybridization (FISH)

The subcellular localization of *ssp1* mRNA in cells expressing Sts5-3xGFP and Dcp1-mCherry was visualized using FISH, and our method was adapted from previously described protocols (*Heinrich et al., 2013*; *Nilsson and Sunnerhagen, 2011*; *Brengues and Parker, 2007*) with the following modifications. Custom Stellaris DNA probes targeted against *ssp1* mRNA were coupled to Quasar 705 (BioSearch Technologies). Cells were fixed with 4% paraformaldehyde for 20 min at room temperature and washed with buffer B (1.2 M sorbitol, 100 mM KHPO$_4$, pH 7.5) Cell walls were digested for 30 min in spheroplast buffer (1.2 M sorbitol, 100 mM KHPO$_4$ at pH 7.5, 20mM vanadyl ribonucleoside complex, 20 µM β-mercaptoethanol) containing 5% Zymolyase 20T at room temperature. Cells were pelleted (taking care to spin cells at ≤500 rpm for 3–5 min between washes in steps after the Zymolyase digestion) then washed in buffer B. Cells were then incubated in 1 mL of -80°C methanol, stored overnight at -20°C, incubated in 1 mL of acetone for 1 min, and then washed twice in 1 mL of 2X SSC (0.3 M NaCl, 30 mM sodium citrate). Cells were preincubated at 37°C in 50 µl of hybridization buffer, consisting of a 1:1 ratio of Buffer F (20% formamide, 10 mM NaHPO4 at pH 7.0) and Buffer H (4X SSC, 4 mg/ml,1 purified BSA and 20mM vanadyl ribonuclease complex) and 2 µl of 10-mg/ml salmon-sperm DNA (which was boiled for 3 min at 95°C). After 1 hr of prehybridization, 0.5 µl of 12.5 µM Quasar 705-conjugated *ssp1* probe was added, and the cells were incubated at 37°C for 5 hr. Cells were washed two times with 2X SSC and resuspended in 2X SSC buffer. Object-based colocalization analysis (based on the distance between centers of mass) was performed using the ImageJ plugin JACoP (Just Another Colocalization Plugin) (*Bolte and Cordelières, 2006*). For threshold selection, we adapted a method similar to one previously described for image threshold selection of RNA FISH images (*Raj et al., 2008*). Specifically, we applied a Laplacian of Gaussian filter to reduce noise and highlight areas of rapid change in each cell and chose the threshold where the histogram reached a plateau, indicating a region where above-background pixels can be clearly detected.

## Glucose depletion in Orb6 overexpressing cells

pREP3X-Orb6- and pREP3X-carrying cells expressing Sts5-3xGFP and Dcp1p-mCherry were grown in absence of thiamine for 18 hr at 32°C. Cells were then washed once in minimal medium minus glucose and resuspended in minimal medium containing 2% or 0% glucose and the required supplements. Cultures were incubated at 32°C for 1 hr before visualizing the localization of Sts5-3xGFP and Dcp1-mCherry using fluorescence microscopy.

## Acknowledgements

We thank Juan Rodriguez for technical support and Dr. Seth Schwartz for help with statistical data analysis. We thank Dr. Mohan Balasubramanian (University of Warwick, Coventry, United Kingdom), Dr. Pilar Perez (University of Salamanca, Salamanca, Spain), and Dr. Eric Chang (Baylor College of Medicine) for providing strains and plasmids. Work in FV's laboratory is supported by the National Institutes of Health R01 grant number GM095867. Part of this work was also supported by NSF grant 0745129. TT was supported by Japan Society for the Promotion of Science grants 16H02503 and 16K14672 and by Cancer Research UK.

## Additional information

### Funding

| Funder | Grant reference number | Author |
|---|---|---|
| National Institutes of Health | GM095867 | Marbelys Rodriguez Pino<br>David J Wiley<br>Maitreyi E Das<br>Chuan Chen<br>Fulvia Verde<br>Illyce Nuñez |
| National Science Foundation | 0745129 | Maitreyi E Das<br>Fulvia Verde |
| Japan Society for the Promotion of Science | 16H02503 | Takashi Toda |
| Japan Society for the Promotion of Science | 16K14672 | Takashi Toda |
| Cancer Research UK | | Takashi Toda |

The funders had no role in study design, data collection and interpretation, or the decision to submit the work for publication.

### Author contributions

IN, Conception and design, Acquisition of data, Analysis and interpretation of data, Drafting or revising the article; MRP, DJW, MED, Conception and design, Acquisition of data, Analysis and interpretation of data; CC, Acquisition of data, Analysis and interpretation of data, Drafting or revising the article; TG, Acquisition of data, Contributed unpublished essential data or reagents; KK, Acquisition of data, Drafting or revising the article, Contributed unpublished essential data or reagents; DH, TT, Drafting or revising the article, Contributed unpublished essential data or reagents; FV, Conception and design, Analysis and interpretation of data, Drafting or revising the article

### Author ORCIDs

Fulvia Verde, http://orcid.org/0000-0002-2575-0823

## Additional files

**Supplementary files**
• Supplementary file 1. List of strains used in this study.

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
