## [Decision Letter]

[Editors’ note: this article was originally rejected after discussions between the reviewers, but the authors were invited to resubmit after an appeal against the decision.]

Thank you for submitting your work entitled "Spatial control of translation repression and polarized growth by conserved NDR kinase Orb6 and RNA-binding protein Sts5" for consideration by *eLife*. Your article has been reviewed by three peer reviewers, including Aaron Gitler, and the evaluation has been overseen by J. Paul Taylor as Reviewing Editor and James Manley as the Senior Editor. Our decision has been reached after consultation between the reviewers. Based on these discussions and the individual reviews below, we regret to inform you that the current submission of your work cannot be considered further for publication in *eLife*, at least in its current form.

Please be aware that the Senior Editor, the Reviewing Editor and the 3 additional referees deliberated extensively over this submission. The mechanism described – phosphorylation-dependent control of RNA-binding protein assembly with P-bodies, with consequential spatial and temporal regulation of the expression of a subset of mRNAs – is of great interest to us. The major weakness is insufficient characterization of mRNA interactions with Sts5, demonstration of spatial redistribution of these mRNAs to P-bodies, and correlation with changes in mRNA stability. These concerns, in addition to those elaborated by the reviewers below, would require more extensive additional work than we typically allow for in a revision. We request that revised manuscripts be returned within 2 months. That said, if you are able to address these concerns fully, we would encourage submitting a new manuscript to *eLife*.

Reviewer #1:

Non membrane-delimited subcellular compartments play critical cellular functions essential for life. These include P-bodies and stress granules (SGs), which are composed of RNAs and RNA-binding proteins. Many of the protein constituents of SGs and P-bodies contain domains resembling yeast prions (called prion-like domains or low-complexity domains). These domains afford the protein a remarkable ability to undergo phase transitions. There has been an emergence of interest in this phenomenon since it has broad implications on gene regulation and many other facets of biology.

This new manuscript by Verde and colleagues presents a very exciting discovery showing how a similar phenomenon can be used to regulate polarized cell growth. They show that the fission yeast protein Sts5 forms RNA granule like accumulations during mitosis and this is negatively regulated by a kinase, Orb6. Remarkably, the aggregation of Sts5 serves to sequester certain key polarized growth protein encoding mRNAs. This is a novel example of how regulated aggregation can be harnessed by cell biology to regulate a fundamental biological process.

The authors have combined high quality imaging, biochemistry, and mRNA analyses to provide compelling evidence to support this novel cellular function for Orb6 and Sts5. In a broader way, they have opened up the possibility that similar phase separations can be harnessed in biology to regulate many other fundamental processes.

In my opinion, this manuscript will be of interest to the readers of *eLife* and will make an important conceptual advance. There is only one area that I suggest the authors consider:

Several recent studies have demonstrated that some RNA-binding proteins, such as hnRNPA1 and FUS can undergo liquid-liquid demixing *in vitro* (e.g., Patel et al., Cell 2015; Molliex et al., Cell 2015; Lin et al., Mol Cell 2015). I suggest the authors perform experiments *in vitro* to see if Sts5 can form similar liquid compartments and, if so, if the phosphorylation by Orb6 alters these properties. One part of this story that seems less well developed is the precise role of Orb6 in negatively regulating Sts5 aggregation. The *in vitro* experiments with defined components might help to better define this mechanism.

Reviewer #2:

This manuscript presents a novel (to my knowledge) study into links between polarized growth and mRNA regulation in *S. pombe*. The authors provide both genetic and biochemical evidence for negative regulation of the RNA-binding protein, Sts5, by the NDR kinase, Orb6. Orb6 inhibits Sts5 assembly into puncta, the association with P-bodies, and degradation of Sts5 target mRNAs. They show that Orb6 acts through regulating Sts5's interaction with Rad24.

This work describes a new role for Orb6 kinase in the control of polarized growth independent of Cdc42 pathways. The broad interest and novelty of the work are related to the molecular regulation of RNP body assembly and function by a conserved kinase and the ability to see this regulation highly spatially controlled in the cytoplasm. Below are suggested experiments that could solidify the links in the model but overall I am in favor of publishing this manuscript.

1) The evidence of translational control is limited to a single protein (Figure 3). The authors present evidence that Orb6 inhibits Sts5-association with P-bodies. This in turn promotes stability and perhaps translation of Sts5 bound mRNAs encoding polarity factors. Confidence in the generality of this mechanism would be increased levels of other putative proteins were measured and if Ssp1 was more deeply analyzed. Are Ssp1 protein levels changing in a different mutant that cannot form P-bodies? Do Ssp1 levels change during cell stress and cell division when Sts5 is clearly forming cytoplasmic puncta?

2) Are the mRNAs identified by microarray as controlled by Sts5 changing localization consistent with the model of spatial control of repression/degradation/translation? It would be good to have some other validation of mRNA regulation beyond just the microarrays. Would it be possible to try single molecular RNA FISH and see how distribution changes from clustered as would be if in large RNPs or distributed throughout cytosol? This tool would allow the visualization of local changes in localization of Sts5 mRNA targets in response to Orb6 inhibition or cellular stress and help distinguish, potentially, between the functions of the mitotic Sts5 puncta and P-bodies.

3) Is Sts5 an in vivo target of Orb6 in cells? The results suggest that Orb6-dependent phosphorylation of Sts5 regulates its association with P-bodies or Rad24. Although the authors present an *in vitro* kinase assay result to demonstrate that Orb6 can phosphorylate Sts5, further evidence of Sts5 as a direct substrate in vivo would be more convincing. This in turn would allow the generation of mutants that are not phosphorylatable to actually demonstrate that phosphorylation is the trigger of disassembly of cytoplasmic puncta. Alternatively, does Sst5 form "aggregates" *in vitro* and are these disassembled by Orb6 kinase?

4) Does Orb6 overexpression cause an inability of the cell to react properly to stress?

Reviewer #3:

In this manuscript, the authors show that the lethality of the NDR kinase Orb6 can be suppressed by loss of function in the RNA-binding protein Sts5. They show that Sts5 is phosphorylated by Orb6 kinase, which promotes its association with 14-3-3; forms aggregates upon starvation or upon loss of Orb6 activity; and associates with several mRNAs, including *ssp1* mRNA. Sts5 loss of function leads to increased levels of these and other mRNAs, indicating Ssp5 promotes mRNA degradation. They propose that Orb6 phosphorylation of Sts5 promotes its association of 14-3-3 to antagonize its association with P-bodies, thus preventing the degradation of key mRNA for polarized growth. They further propose that this regulation is spatially regulated, with Sts5 aggregates forming preferentially at the non-growing cell end of monopolar cells; and temporally regulated, with Orb6-mediated inhibition taking place during mitosis and cytokinesis but being relieved upon contractile ring closure.

The ideas put forward by this study are interesting and largely novel. However, I find that several parts of the model are not well supported by the data shown, with some inconsistencies. Several key points are overstated in the text relative to the data shown, which in aggregate fail to convince me of the whole narrative. These are detailed below.

Genetic interaction of *sts5* and *orb6* (Figure 1 and Figure 1—figure supplement 1):

The authors identify *sts5* loss-of-function as suppressor of *orb6* mutation lethality and show some evidence that this suppression is separable from the role of Orb6 in regulating Cdc42 activity. The suppression data is convincing, but the separation of function is less so. In the growth assay shown in Figure 1—figure supplement 2, single colonies cannot be seen in the higher dilutions and the *gef1∆* single mutant strain appears very sick. From this assay, one could argue that the *gef1∆ orb6-as* double mutant actually grows better than the single mutant. This should be clarified. In addition, if the *orb6-sts5* control of growth is really genetically separable from the spatial control of Cdc42, this would predict that Sts5 loss of function does not correct the round shape of *orb6* mutants (at permissive temperature) because of Cdc42 activity presence on cell sides, i.e. *sts5-276 orb6-as* double mutant treated with PP1 inhibitor at 25°C should be round. Is this the case?

I am not sure I understand exactly what is quantified in Figure 1—figure supplement 1. In Figure 1, it is shown that 60% of *orb6-as2* cells are septated. In Figure 1—figure supplement 1, only non-septated cells are shown with CRIB signal on cell sides, with quantification that amounts to 80%. Clearly 60+80>100, so I am not sure how to reconcile these numbers. If only a subset of cells is quantified, this should be specified. Also, in Figure 1, the authors indicate a standard deviation from only n=2 experiments, which cannot be derived.

Formation of Sts5 punctae, association with P-bodies, mRNA degradation and regulation by Orb6:

The authors show that starvation or Orb6 kinase inhibition leads to formation of Sts5 punctae that colocalize with P-body marker Dcp1. Sts5 loss of function also leads to a reduction in the number of P-bodies after Orb6 inhibition, implicating Sts5 in the formation of P-bodies. However, they do not examine the role of Sts5 in P-body formation upon starvation, when P-body structures are much more prominent. Does Sts5 also play a role in the formation of P-bodies upon starvation? During N-starvation, it looks like there is only a partial colocalization of Sts5 with Dcp1. This should be stated in the text.

Regarding mRNA levels, the data on *ssp1* mRNA level regulation is convincing, with epistasis of Sts5 over Orb6, indicating that Orb6 prevents Sts5-mediated degradation of Ssp1 mRNA. However, epistasis is not shown for any of the other transcripts examined. This would be required for the authors to make their claim of a general finding.

The link between aggregate formation, P-bodies and mRNA degradation is also not clear. The authors state, in the Results section, subsection “Orb6 kinase inhibits Sts5 aggregation and Sts5 localization to P-bodies”, that "Orb6 kinase prevents [...] Sts5 localization to P-bodies, with the effect of preventing the degradation of specific mRNAs" and, in the sixth paragraph of the Discussion section, that "Sts5 promotes mRNA degradation likely by recruiting transcripts to processing (P) bodies". However, the observed increased level of several mRNAs in the *sts5∆* strain does not correlate with the colocalization of Sts5 with P-bodies: the latter is shown upon starvation, whereas the mRNA levels are examined in normal growth conditions. Similarly, in *rad24∆* where there is no obvious colocalization of Sts5 with P-body markers, the authors claim Ssp1 levels are reduced (whether this is actually the case is also an issue – see below). From this, it appears that formation of aggregates and mRNA level regulation are not causally linked. It is equally possible that aggregation is a consequence of the *sts5* interaction with/degradation of mRNAs or that the two effects or not directly causally related.

The kinase assay in Figure 6 lacks a negative control to state that Orb6 kinase phosphorylates Sts5. What is shown is that a Mob2-associated kinase phosphorylates Sts5. This may indeed be Orb6, but should be accurately described in the text.

Kinase assay and 14-3-3 role:

This part is not very convincing with text and figures showing a number of inconsistencies and over-interpretations.

The authors state (Results section, subsection “14-3-3 protein Rad24 negatively regulates Sts5 aggregation”) that "*ssp1* mRNA levels are consistently reduced in *rad24∆* mutants", but the data shown in Figure 6 does not state a statistically significant difference between wt and *rad24∆* strains. The fact that the levels are significantly reduced in *rad24∆* compared to *sts5∆* or *sts5∆ rad24∆* double mutants does not indicate anything about the possible role of Rad24, simply that Sts5 promotes Ssp1 degradation.

Similarly, the authors propose on later in the Results section that "Orb6 inhibits [...] Sts5 localization to P-bodies by promoting its interaction with 14-3-3 protein Rad24." However, the data shows that Sts5 forms aggregates in *rad24∆* cells, but I do not see evidence that they contain Dcp1 P-body marker (see Figure 6). The authors indeed actually write that P-body formation is not strongly induced in *rad24∆*.

Temporal regulation:

The proposed anti-correlation of Ssp5 aggregates with Orb6 activity during cell division is interesting, but is poorly supported. Previous work had shown that Orb6 activity is inhibited by SIN signaling until closure of the contractile ring. Here, the authors rely solely on this published evidence to propose a causal link between Ssp5 aggregate dissipation during septation and Orb6 activity. They should at least precisely quantify the number of aggregates over time during cytokinesis. For this a timelapse acquisition of Sts5-3xGFP would be best, relative to a cytokinesis marker to time the progression of ring constriction. They should also show that any disappearance of Sts5 aggregates is indeed due to Orb6 activity, using either of their *orb6-as* or *orb6* ts alleles.

The localization of Orb6-GFP during cell division, claimed to be shown in Figure 7—figure supplement 2C, is not shown. That figure legend is also present at the bottom of Figure 7—figure supplement 1.

The cell lysis phenotype of *sts5∆* cells should be quantified.

Spatial regulation:

In general, the spatial regulation of this mRNA stabilization control is not very convincing. The asymmetry in the aggregates of Sts5 is rather difficult to see. It is well documented in *tea1∆* cells, but less so in wt cells. Could the authors provide more examples to show the Sts5 asymmetric localization in monopolar wt cells? The authors propose the existence of an Sts5 phosphorylation gradient. This is an interesting idea, with conceptual similarity to gradients of MEX-5 in *C. elegans* that contribute to the dissolution of P-granules, but is it actually possible at the small scale of a yeast cell? Diffusion rates in the cytosol, if Sts5 diffuses unimpeded, makes this unlikely, but it could be constrained through aggregate formation. Examination of the motility of Sts5 aggregates would help to resolve this point.

I am also not convinced that Sts5 has a direct role in regulating bipolar growth. As the *sts5∆* mutant cells are longer at all densities examined, the increased proportion of bipolar cells could simply be a result of the longer cell length. It is well established that fission yeast cells have a cell size-dependent regulation of bipolarity. The authors instead seem to argue that the increased length is a consequence of increased bipolarity, but I do not see evidence for this. For cells of equal initial length, are the *sts5∆* mutant cells growing faster? Are cells of same length more bipolar if *sts5∆*?

Allele nomenclature and temperature sensitivity:

The nomenclature of the alleles used is confusing. It should be clearly stated in the text that *orb4* and *sts5* are two names for the same gene (and as much as possible use a single gene name throughout the manuscript), and that the alleles used are both early stop codons. The authors also use a deletion of the gene, but it is not very clear which strain is used in which experiment, and why different strains were used. For instance, in second paragraph of Results section, the authors say they "examined whether *sts5* deletion also suppresses the loss of polarity observed upon Orb6 kinase inhibition", but the figure referred to (Figure 1—figure supplement 1) states use of *orb4-A9* allele.

More generally, I do not understand the use of different "permissive", "semi-permissive" and "restrictive" temperatures to examine the phenotype of *sts5* loss of function. In what way are the alleles used conditional if these are either early stop codons or deletions? This should be clearly stated and discussed. For instance, the increased levels of Ssp1-HA in *sts5* mutants is only shown to happen at 36°C, but the text states that the increased levels are "exaggerated" at 36°C (Results section, subsection “Loss of Sts5 leads to increased levels of mRNAs involved in growth control and bipolar growth activation”), which implies it is also the case at lower temperature.

[Editors’ note: what now follows is the decision letter after the authors submitted for further consideration.]

Thank you for submitting your article "Spatial control of translation repression and polarized growth by conserved NDR kinase Orb6 and RNA-binding protein Sts5" for consideration by *eLife*. Your article has been reviewed by three peer reviewers, including Aaron D Gitler (Reviewer #1), and the evaluation has been overseen by a Reviewing Editor and James Manley as the Senior Editor.

The reviewers have discussed the reviews with one another and the Reviewing Editor has drafted this decision to help you prepare a revised submission. We hope you will be able to submit the revised version within two months, so please let us know if you have any questions first.

Summary:

This manuscript presents novel insight into a link between mRNA regulation and polarized growth in *S. pombe*. The authors present genetic and biochemical evidence indicating that NDR kinase Orb6 negatively regulates the RNA-binding protein Sts5. Orb6 activity alters an interaction of Sts5 with 14-3-3, regulating the ability of Sts5 to assemble into cytoplasmic P-bodies, which sequester and influence the stability of Sts5 target mRNAs. There was consensus among the reviewers that the additional data and analyses provided in the revised manuscript addressed most of the concerns from the original review, but several issues remain that must be addressed.

Essential revisions:

1. The FISH experiment showing colocalization of *ssp1* mRNA with Sts5 (and Dcp1) granules is a nice addition. However, in addition to showing representative images the extent of colocalization should quantified.

2. Similarly, the analysis of Efc25 protein levels is a nice addition, though it needs to be quantified. An epistasis experiment to test whether deletion of *sts5* abrogates the reduction in Efc25 levels observed in *orb6* mutant (as shown for Ssp1) would further support the conclusion.

3. In Figure 6, the statistical comparisons should be done between *rad24∆* and wt, not between *rad24∆* and *sts5∆*.

---

## [Author Response]

[Editors’ note: the author responses to the first round of peer review follow.]

Thank you for submitting your work entitled "Spatial control of translation repression and polarized growth by conserved NDR kinase Orb6 and RNA-binding protein Sts5" for consideration by eLife. Your article has been reviewed by three peer reviewers, including Aaron Gitler, and the evaluation has been overseen by J. Paul Taylor as Reviewing Editor and James Manley as the Senior Editor. Our decision has been reached after consultation between the reviewers. Based on these discussions and the individual reviews below, we regret to inform you that the current submission of your work cannot be considered further for publication in eLife, at least in its current form.

*Please be aware that the Senior Editor, the Reviewing Editor and the 3 additional referees deliberated extensively over this submission. The mechanism described – phosphorylation-dependent control of RNA-binding protein assembly with P-bodies, with consequential spatial and temporal regulation of the expression of a subset of mRNAs – is of great interest to us. The major weakness is insufficient characterization of mRNA interactions with Sts5, demonstration of spatial redistribution of these mRNAs to P-bodies, and correlation with changes in mRNA stability. These concerns, in addition to those elaborated by the reviewers below, would require more extensive additional work than we typically allow for in a revision. We request that revised manuscripts be returned within 2 months. That said, if you are able to address these concerns fully, we would encourage submitting a new manuscript to eLife.*

Reviewer #1:

*[…] In my opinion, this manuscript will be of interest to the readers of eLife and will make an important conceptual advance. There is only one area that I suggest the authors consider:*

*Several recent studies have demonstrated that some RNA-binding proteins, such as hnRNPA1 and FUS can undergo liquid-liquid demixing* in vitro *(e.g., Patel et al., Cell 2015; Molliex et al., Cell 2015; Lin et al., Mol Cell 2015). I suggest the authors perform experiments* in vitro *to see if Sts5 can form similar liquid compartments and, if so, if the phosphorylation by Orb6 alters these properties. One part of this story that seems less well developed is the precise role of Orb6 in negatively regulating Sts5 aggregation. The* in vitro *experiments with defined components might help to better define this mechanism.*

Sts5 protein contains a predicted unstructured N terminal domain in the first 301 amino acids. We have included this finding in the current version of the paper (Figure 2—figure supplement 1). The role of unstructured domains in the formation of liquid compartments and cytoplasmic phase transition is a biological problem of fundamental importance, and we plan to devote substantial effort in the future to reconstitute the behavior of Sts5 in vitro. At the moment however, we feel these experiments are beyond the scope of this paper, and it would require an entire additional manuscript to present them completely and well controlled. Also, the phosphorylation studies are complicated by the existence of other phosphorylated sites in the Sts5 molecule by which other kinases may contribute to its regulation (results of proteomic analysis: Wilson-Grady et al., 2008; Koch et al., 2011; Carpy et al., 2014; Kettenbach et al., 2015), and that may, at this time, confound the in vitro experiments.

*Reviewer #2:*

*[…] This work describes a new role for Orb6 kinase in the control of polarized growth independent of Cdc42 pathways. The broad interest and novelty of the work are related to the molecular regulation of RNP body assembly and function by a conserved kinase and the ability to see this regulation highly spatially controlled in the cytoplasm. Below are suggested experiments that could solidify the links in the model but overall I am in favor of publishing this manuscript.*

*1) The evidence of translational control is limited to a single protein (Figure 3). The authors present evidence that Orb6 inhibits Sts5-association with P-bodies. This in turn promotes stability and perhaps translation of Sts5 bound mRNAs encoding polarity factors. Confidence in the generality of this mechanism would be increased levels of other putative proteins were measured and if Ssp1 was more deeply analyzed. Are Ssp1 protein levels changing in a different mutant that cannot form P-bodies? Do Ssp1 levels change during cell stress and cell division when Sts5 is clearly forming cytoplasmic puncta?*

We further characterized Efc25, the protein encoded by Sts5-bound *efc25* mRNA, whose levels increase in *sts5Δ* cells and decrease in *orb6* mutant cells. We found that Efc25 protein levels increase in *sts5Δ* cells and decrease in *orb6* mutants, similarly to Ssp1 protein (see Figure 3—figure supplement 1; Figure 5—figure supplement 1E).

As far as Ssp1 protein levels in mutants that cannot form P-bodies areconcerned, we have tested Ssp1 levels in *pdc1Δ* and *edc3Δ* mutant cells that encode Pbody components that regulate mRNA decapping. We found that Ssp1 protein levels are increased in the *pdc1Δ* and *edc3Δ* backgrounds (see Figure 3).

*2) Are the mRNAs identified by microarray as controlled by Sts5 changing localization consistent with the model of spatial control of repression/degradation/translation? It would be good to have some other validation of mRNA regulation beyond just the microarrays. Would it be possible to try single molecular RNA FISH and see how distribution changes from clustered as would be if in large RNPs or distributed throughout cytosol? This tool would allow the visualization of local changes in localization of Sts5 mRNA targets in response to Orb6 inhibition or cellular stress and help distinguish, potentially, between the functions of the mitotic Sts5 puncta and P-bodies.*

Consistent with Sts5 controlling the localization of *ssp1* mRNA, we found that *ssp1* fluorescent RNA FISH probes (Stellaris) colocalize with a subset of Sts5-GFP and Dcp1- mCherry containing punta in glucose-starved cells (see Figure 3 and Figure 3—figure supplement 1 D). Our results are consistent with experiments published by other investigators, for example by the laboratory of Roy Parker, whose protocol we adapted (Brengues and Parker, 2007).

*3) Is Sts5 an* in vivo *target of Orb6 in cells? The results suggest that Orb6-dependent phosphorylation of Sts5 regulates its association with P-bodies or Rad24. Although the authors present an* in vitro *kinase assay result to demonstrate that Orb6 can phosphorylate Sts5, further evidence of Sts5 as a direct substrate* in vivo *would be more convincing. This in turn would allow the generation of mutants that are not phosphorylatable to actually demonstrate that phosphorylation is the trigger of disassembly of cytoplasmic puncta. Alternatively, does Sst5 form "aggregates"* in vitro *and are these disassembled by Orb6 kinase?*

Sts5 protein contains an unstructured N-terminal domain comprising the first 301 amino acids (Figure 2—figure supplement 1). However it is not recruited into RNPs in the absence of the mRNA-binding domain. The Sts5 sequence presents several canonical Orb6 consensus sites and other lower affinity Orb6 consensus sites that may also need to be modified to ensure maximal response. Also, the phosphorylation studies are complicated by the existence of other phosphorylated sites by other kinases in different domains of the Sts5 molecule that may contribute to its regulation (results of proteomic analysis (Wilson-Grady et al., 2007; Koch et al., 2011; Carpy et al., 2014; Kettenbach et al., 2015), and that may confound in vitro experiments. These are interesting experiments however, and are going to be the subject of extensive efforts in the lab in the future.

*4) Does Orb6 overexpression cause an inability of the cell to react properly to stress?*

Nutritional stress induces Sts5 recruitment to RNPs and P-body formation. Thus, we used Sts5 recruitment to RNPs and P-body formation as a read-out of stress. We have over-expressed Orb6 kinase and monitored P-body formation and Sts5 recruitment to RNPs upon glucose deprivation. We found that Orb6 overexpression significantly decreases the intensity of Sts5 puncta and P-bodies (see Figure 5— figure supplement 2A).

*Reviewer #3:*

*In this manuscript, the authors show that the lethality of the NDR kinase Orb6 can be suppressed by loss of function in the RNA-binding protein Sts5. They show that Sts5 is phosphorylated by Orb6 kinase, which promotes its association with 14-3-3; forms aggregates upon starvation or upon loss of Orb6 activity; and associates with several mRNAs, including ssp1 mRNA. Sts5 loss of function leads to increased levels of these and other mRNAs, indicating Ssp5 promotes mRNA degradation. They propose that Orb6 phosphorylation of Sts5 promotes its association of 14-3-3 to antagonize its association with P-bodies, thus preventing the degradation of key mRNA for polarized growth. They further propose that this regulation is spatially regulated, with Sts5 aggregates forming preferentially at the non-growing cell end of monopolar cells; and temporally regulated, with Orb6-mediated inhibition taking place during mitosis and cytokinesis but being relieved upon contractile ring closure.*

*The ideas put forward by this study are interesting and largely novel. However, I find that several parts of the model are not well supported by the data shown, with some inconsistencies. Several key points are overstated in the text relative to the data shown, which in aggregate fail to convince me of the whole narrative. These are detailed below.*

*Genetic interaction of sts5 and orb6 (Figure 1 and Figure 1—figure supplement 1):*

*The authors identify sts5 loss-of-function as suppressor of orb6 mutation lethality and show some evidence that this suppression is separable from the role of Orb6 in regulating Cdc42 activity. The suppression data is convincing, but the separation of function is less so. In the growth assay shown in Figure 1—figure supplement 2, single colonies cannot be seen in the higher dilutions and the gef1∆ single mutant strain appears very sick. From this assay, one could argue that the gef1∆ orb6-as double mutant actually grows better than the single mutant. This should be clarified.*

The growth delay of the *gef1Δ* strain observed in Figure 1—figure supplement 2, as compared to *gef1Δ orb6-as2*, in the control DMSO condition, was due to a difference in the background of the strains. The *gef1Δ* strain was burdened by *ura4Δ* and *leu1-32* auxotrophic markers, which produced a growth difference even in the presence of exogenously added supplements. We have repeated these experiments with a different *gef1Δ* strain that does not contain these mutations, and found no difference in growth between *gef1Δ* and *gef1Δ orb6-as2* in DMSO. (See Figure 1—figure supplement 2)

*In addition, if the orb6-sts5 control of growth is really genetically separable from the spatial control of Cdc42, this would predict that Sts5 loss of function does not correct the round shape of orb6 mutants (at permissive temperature) because of Cdc42 activity presence on cell sides, i.e. sts5-276 orb6-as double mutant treated with PP1 inhibitor at 25*°*C should be round. Is this the case?*

Indeed, we have analyzed *sts5Δ orb6-as2* double mutant cells, treated with 1- Na-PP1 inhibitor at 25°C for 5 hours, and found them to be round. We show this experiment in the current version of the paper (see Figure 1—figure supplement 1)

*I am not sure I understand exactly what is quantified in Figure 1—figure supplement 1. In Figure 1, it is shown that 60% of orb6-as2 cells are septated. In Figure 1—figure supplement 1, only non-septated cells are shown with CRIB signal on cell sides, with quantification that amounts to 80%. Clearly 60+80>100, so I am not sure how to reconcile these numbers. If only a subset of cells is quantified, this should be specified.*

In Figure 1—figure supplement 1 we quantified the extent of CRIB-GFP mis-localization only in non-septating cells (cells that do not contain a septum, which indeed is a subset). We have clarified this in the corresponding legend of the Figure 1—figure supplement 1. Also, ectopic localization of CRIB-GFP is a rapid response that can be observed as early as 15 minutes after Orb6-as2 inhibition with 1-NA-PP1 (Das et al., 2009). Thus, the experimental conditions in the experiment depicted in Figure 1—figure supplement 1 and 1D are different than in Figure 1, in which we quantified septation index after prolonged Orb6- as2 inhibition (2 hours) and the proportion of septated cells was different in the two experiments.

*Also, in Figure 1, the authors indicate a standard deviation from only n=2 experiments, which cannot be derived.*

As also mentioned below, we counted almost 400 cells for each condition, giving us confidence in our conclusion. The reviewer, however, is correct to point out the issue with the standard deviation. In the revised version, we show in Figure 1 an entirely different set of three experiments using the *sts5Δ* strain, instead of the *orb4-A9* allele. This was done in response to point 6 of this reviewer, to simplify the use of different alleles. The results are entirely consistent with the previous experiment, and the 6 experiments were performed three times (see Figure 1).

*Formation of Sts5 punctae, association with P-bodies, mRNA degradation and regulation by Orb6:*

*The authors show that starvation or Orb6 kinase inhibition leads to formation of Sts5 punctae that colocalize with P-body marker Dcp1. Sts5 loss of function also leads to a reduction in the number of P-bodies after Orb6 inhibition, implicating Sts5 in the formation of P-bodies. However, they do not examine the role of Sts5 in P-body formation upon starvation, when P-body structures are much more prominent. Does Sts5 also play a role in the formation of P-bodies upon starvation?*

As mentioned above, to address the role of Sts5 during nutritional stress we have performed additional experiments that were not included in the original version of the paper. In the current version, we show that loss of *sts5 (sts5Δ*) significantly decreases the intensity of P-bodies formed following glucose depletion. We added these experiments to the supplement section of the paper (Figure 5—figure supplement 2 B-D).

*During N-starvation, it looks like there is only a partial colocalization of Sts5 with Dcp1. This should be stated in the text.*

We have mentioned this fact in the text (Results section, subsection “Sts5 proteins are recruited into cytoplasmic puncta during mitosis and during nutritional starvation”).

*Regarding mRNA levels, the data on ssp1 mRNA level regulation is convincing, with epistasis of Sts5 over Orb6, indicating that Orb6 prevents Sts5-mediated degradation of Ssp1 mRNA. However, epistasis is not shown for any of the other transcripts examined. This would be required for the authors to make their claim of a general finding.*

As mentioned above, we now characterize also *efc25*, another mRNA that copurifies with Sts5 protein, whose levels increase in *sts5Δ* cells and decrease in *orb6* mutant cells. Similarly to Ssp1 protein, Efc25 protein levels increase in *sts5Δ* cells and decrease in *orb6* mutants (see Figure 3—figure supplement 1; Figure 5—figure supplement 1, E).

*The link between aggregate formation, P-bodies and mRNA degradation is also not clear. The authors state, in the Results section, subsection “Orb6 kinase inhibits Sts5 aggregation and Sts5 localization to P-bodies”, that "Orb6 kinase prevents [...] Sts5 localization to P-bodies, with the effect of preventing the degradation of specific mRNAs" and, in the sixth paragraph of the Discussion section, that "Sts5 promotes mRNA degradation likely by recruiting transcripts to processing (P) bodies". However, the observed increased level of several mRNAs in the sts5∆ strain does not correlate with the colocalization of Sts5 with P-bodies: the latter is shown upon starvation, whereas the mRNA levels are examined in normal growth conditions. Similarly, in rad24∆ where there is no obvious colocalization of Sts5 with P-body markers, the authors claim Ssp1 levels are reduced (whether this is actually the case is also an issue – see below). From this, it appears that formation of aggregates and mRNA level regulation are not causally linked. It is equally possible that aggregation is a consequence of the sts5 interaction with/degradation of mRNAs or that the two effects or not directly causally related.*

Indeed, our data indicates that in the absence of Sts5, levels of certain Sts5- associated mRNAs are increased even in the absence of extensive or visible P-body formation. Previous findings in *S. cerevisiae* indicate that mRNA degradation proceeds to a certain extent even in the absence of substantial P-body formation. Processes such as mRNA decay and RNA-mediated gene silencing (Eulalio et al., 2007) and mRNA 7 decapping and translational repression (Decker et al., 2007) are functional in cells lacking detectable microscopic P-bodies indicating that even though P-body components play crucial roles in mRNA silencing and decay, localization of mRNAs into large P-bodies is not necessary for function. Consistent with this idea, we find that loss of P-body components Pdc1 and Edc3 leads to a significant increase in Ssp1 protein levels, even in the absence of stress, when P-bodies are not visible (see Figure 3).

As the reviewer mentions, our data also show that Sts5 recruitment into large RNPs in the *rad24Δ* strain does not trigger a substantial decrease in *ssp1* mRNA levels. Ssp1 mRNA levels, however, are decreased upon Orb6 kinase inhibition, which induces both P-body formation and Sts5 protein recruitment, in addition to other effects, such as Cdc42 activation. Thus, our data suggest that additional yet unknown signals, dependent on Orb6 kinase, are required for inducing mRNA degradation. We have clarified our conclusions in the current version of the paper (e.g. Results section, subsection “14-3-3 protein Rad24 negatively regulates Sts5 recruitment into cytoplasmic puncta”; fourth paragraph of Discussion section).

*The kinase assay in Figure 6 lacks a negative control to state that Orb6 kinase phosphorylates Sts5. What is shown is that a Mob2-associated kinase phosphorylates Sts5. This may indeed be Orb6, but should be accurately described in the text.*

We have changed the text to reflect this caveat (Results section, subsection “14-3-3 protein Rad24 negatively regulates Sts5 recruitment into cytoplasmic puncta”).

*Kinase assay and 14-3-3 role:*

*This part is not very convincing with text and figures showing a number of inconsistencies and over-interpretations.*

*The authors state (Results section, subsection “14-3-3 protein Rad24 negatively regulates Sts5 aggregation”) that "ssp1 mRNA levels are consistently reduced in rad24∆ mutants", but the data shown in Figure 6 does not state a statistically significant difference between wt and rad24∆ strains. The fact that the levels are significantly reduced in rad24∆ compared to sts5∆ or sts5∆ rad24∆ double mutants does not indicate anything about the possible role of Rad24, simply that Sts5 promotes Ssp1 degradation.*

*Similarly, the authors propose on later in the Results section that "Orb6 inhibits.… Sts5 localization to P-bodies by promoting its interaction with 14-3-3 protein Rad24." However, the data shows that Sts5 forms aggregates in rad24∆ cells, but I do not see evidence that they contain Dcp1 P-body marker (see Figure 6). The authors indeed actually write that P-body formation is not strongly induced in rad24∆.*

Our conclusion from experiments detailed in Figure 6 is that one of the roles of Orb6 kinase is to regulate association of Sts5 with Rad24 and prevent Sts5 recruitment into RNPs. Our experiments with *rad24Δ* mutant cells indeed show that Sts5 recruitment into RNPs occurs independently of substantial P-body formation, since these cells are not starved. We tried to be clearer in the current version of the paper. We speculate from our findings and propose that Sts5 protein regulation by Orb6 kinase is part of a mechanism ultimately regulating mRNA association with the P-bodies, since we see P-body formation upon Orb6 kinase inhibition. We also rephrased this part.

*Temporal regulation:*

*The proposed anti-correlation of Ssp5 aggregates with Orb6 activity during cell division is interesting, but is poorly supported. Previous work had shown that Orb6 activity is inhibited by SIN signaling until closure of the contractile ring. Here, the authors rely solely on this published evidence to propose a causal link between Ssp5 aggregate dissipation during septation and Orb6 activity. They should at least precisely quantify the number of aggregates over time during cytokinesis. For this a timelapse acquisition of Sts5-3xGFP would be best, relative to a cytokinesis marker to time the progression of ring constriction. They should also show that any disappearance of Sts5 aggregates is indeed due to Orb6 activity, using either of their orb6-as or orb6 ts alleles.*

We have quantified the number of Sts5 puncta during cytokinesis in the presence or absence of Orb6 kinase. As a marker for cytokinesis we used Rlc1, in addition to calcofluor. Rlc1, myosin II regulatory light chain (Le Goff et al., 2000) (Wu et al., 2006), disappears from the site of division when the actomyosin ring has fully contracted and the septum is closed (Wei, 2016). We found that Sts5 puncta are present as long as Rlc1 is present at the site of division (a sign that septum closure in not complete) but disappear in cells where Rlc1 has left the site of division, after septum closure. Indicating that disappearance of Sts5 puncta is indeed due to Orb6 activity, we find that Sts5 puncta are still visible in cells that have completed septum closure (and where Rlc1 has left the site of division), when Orb6 activity is inhibited. We show these experiments in the current version of the paper (see Figure 7; Figure 7—figure supplement 2).

*The localization of Orb6-GFP during cell division, claimed to be shown in Figure 7—figure supplement 2, is not shown. That figure legend is also present at the bottom of Figure —figure supplement 1.*

We indeed showed the localization of Orb6-GFP at the septum in Figure 7—figure supplement 1 (c), as the reviewer mentions. Figure 7—figure supplement 2C was a single typo, for which we apologize, as it should have read Figure 7—figure supplement 1 (c). We corrected this typo in the text.

*The cell lysis phenotype of sts5∆ cells should be quantified.*

We quantified the lysis phenotype of *sts5Δ* in triplicate experiments, and found that, under our experimental conditions, 43% of septating *sts5Δ* cells lyse at 36*°* C, as compared to 8% septating control cells. We added this information in the corresponding legend (see Figure 7—figure supplement 2).

*Spatial regulation:*

*In general, the spatial regulation of this mRNA stabilization control is not very convincing. The asymmetry in the aggregates of Sts5 is rather difficult to see. It is well documented in tea1∆ cells, but less so in wt cells. Could the authors provide more examples to show the Sts5 asymmetric localization in monopolar wt cells? The authors propose the existence of an Sts5 phosphorylation gradient. This is an interesting idea, with conceptual similarity to gradients of MEX-5 in C.elegans that contribute to the dissolution of P-granules, but is it actually possible at the small scale of a yeast cell? Diffusion rates in the cytosol, if Sts5 diffuses unimpeded, makes this unlikely, but it could be constrained through aggregate formation. Examination of the motility of Sts5 aggregates would help to resolve this point.*

Indeed, to verify the observation of asymmetric Sts5 puncta in small monopolar cells, first observed in wild-type cells, we used *tea1Δ* cells since they are 9 longer, stay monopolar, and localize Orb6-GFP at only one tip. We have added more examples of small wild-type cells that reflect the average representative localization under conditions of active cell growth (see Figure 7—figure supplement 1). To avoid confusion or overstatements, we removed the word “gradient” that was originally proposed in the discussion.

*I am also not convinced that Sts5 has a direct role in regulating bipolar growth. As the sts5∆ mutant cells are longer at all densities examined, the increased proportion of bipolar cells could simply be a result of the longer cell length. It is well established that fission yeast cells have a cell size-dependent regulation of bipolarity. The authors instead seem to argue that the increased length is a consequence of increased bipolarity, but I do not see evidence for this. For cells of equal initial length, are the sts5∆ mutant cells growing faster? Are cells of same length more bipolar if sts5∆?*

In response to this reviewer’s comments, we have performed two additional sets of experiments. We show that *sts5Δ* cells display increased bipolarity even during exponential cell growth, when *sts5Δ* cells display the same size (average cell length at division; see Figure 8). As this reviewer suggested, we measured the percentage of bipolar cells in cells of similar length (less than 10 micrometers; from 10 to 12; more than 12), and found that *sts5Δ* cells display an increased percentage of bipolar cells as compared to controls (see Figure 8—figure supplement 1) even if *sts5Δ* cells display the same growth rate as control *sts5*+ cells (Figure 8—figure supplement 1).

*Allele nomenclature and temperature sensitivity:*

*The nomenclature of the alleles used is confusing. It should be clearly stated in the text that orb4 and sts5 are two names for the same gene (and as much as possible use a single gene name throughout the manuscript), and that the alleles used are both early stop codons. The authors also use a deletion of the gene, but it is not very clear which strain is used in which experiment, and why different strains were used. For instance, in second paragraph of Results section, the authors say they "examined whether sts5 deletion also suppresses the loss of polarity observed upon Orb6 kinase inhibition", but the figure referred to (Figure 1—figure supplement 1) states use of orb4-A9 allele.*

We have redone the experiments in Figure 1, using the *sts5Δ* allele, and obtained identical results to the *orb4-A9* allele. Aside from growth suppression experiments detailed in Figure 1 (done some time ago with the *sts5-276* allele by our collaborators at Hiroshima University) and one control experiment with CRIB-GFP (which was part of the initial screening with *orb4-A9* allele; Figure 1—figure supplement 1), all other experiments were performed using the *sts5Δ* allele. As the reviewer mentions, these are all early stop codons, and we never found the corresponding mutants to behave differently from the *sts5Δ* deletion mutant cells. We mentioned this aspect in the legend of Figure 1—figure supplement 1.

*More generally, I do not understand the use of different "permissive", "semi-permissive" and "restrictive" temperatures to examine the phenotype of sts5 loss of function. In what way are the alleles used conditional if these are either early stop codons or deletions? This should be clearly stated and discussed. For instance, the increased levels of Ssp1-HA in sts5 mutants is only shown to happen at 36°C, but the text states that the increased levels are "exaggerated" at 36°C (Results section, subsection “Loss of Sts5 leads to increased levels of mRNAs involved in growth control and bipolar growth activation”), which implies it is also the case at lower temperature.*

In the current version of the paper, for clarity, we decided not use “restrictive” or “permissive” definitions and just state the temperature at which experiments are performed. The use of “restrictive” or “permissive” temperature refers to conditional, temperature-sensitive phenotypes, and thus it is a functional definition. Indeed *sts5Δ* displays a morphological phenotype that is temperature-sensitive: it is cylindrical in shape (albeit with an increase in the extent of cell growth at the new end) at 25*°*C, while it is enlarged and round at 36*°*C. These phenotypes likely reflect a decreased ability for *sts5Δ* cells to adapt to higher temperatures. We changed the wording in seventh paragraph, Results section.

[Editors’ note: the author responses to the re-review follow.]

*Thank you for submitting your article "Spatial control of translation repression and polarized growth by conserved NDR kinase Orb6 and RNA-binding protein Sts5" for consideration by eLife. Your article has been reviewed by three peer reviewers, including Aaron D Gitler (Reviewer #1), and the evaluation has been overseen by a Reviewing Editor and James Manley as the Senior Editor.*

*The reviewers have discussed the reviews with one another and the Reviewing Editor has drafted this decision to help you prepare a revised submission. We hope you will be able to submit the revised version within two months, so please let us know if you have any questions first.*

*Summary:*

*This manuscript presents novel insight into a link between mRNA regulation and polarized growth in S. pombe. The authors present genetic and biochemical evidence indicating that NDR kinase Orb6 negatively regulates the RNA-binding protein Sts5. Orb6 activity alters an interaction of Sts5 with 14-3-3, regulating the ability of Sts5 to assemble into cytoplasmic P-bodies, which sequester and influence the stability of Sts5 target mRNAs. There was consensus among the reviewers that the additional data and analyses provided in the revised manuscript addressed most of the concerns from the original review, but several issues remain that must be addressed.*

*Essential revisions:*

*1. The FISH experiment showing colocalization of ssp1 mRNA with Sts5 (and Dcp1) granules is a nice addition. However, in addition to showing representative images the extent of colocalization should quantified.*

We have quantified the co-localization of *ssp1* mRNA and Sts5 (and Dcp1) granules, and added these data in Figure 4—figure supplement 3. We used the ImageJ plugin JACoP (Just Another Colocalization Plugin) to perform object-based quantification based on the distance between centers of mass of puncta (Bolte and Cordelieres, 2006). We measured, respectively, the proportion of *ssp1* puncta colocalizing with Sts5-3xGFP or Dcp1-mCh (Figure 4—figure supplement 3), and the proportion of Sts5-3xGFP or Dcp1-mCh puncta colocalizing with *ssp1* puncta (Figure 4—figure supplement 3), in three independent experiments, in the presence or absence of glucose.

*2. Similarly, the analysis of Efc25 protein levels is a nice addition, though it needs to be quantified. An epistasis experiment to test whether deletion of sts5 abrogates the reduction in Efc25 levels observed in orb6 mutant (as shown for Ssp1) would further support the conclusion.*

We have performed additional experiments, and showed that Efc25 levels decrease in inhibitor sensitive *orb6-as2* mutants (Figure 6—figure supplement 1). We also found that *sts5* deletion abolishes the reduction of Efc25 levels in *orb6-as2* mutants (Figure 6—figure supplement 1). We have quantified the results of three independent experiments in Figure 6—figure supplement 1, right panel.

*3. In Figure 6, the statistical comparisons should be done between rad24∆ and wt, not between rad24∆ and sts5∆.*

We have changed the Figure 7 (Figure 8 in the current manuscript)appropriately to reflect the message of the experiment, that there is not a significant difference in *ssp1* mRNA levels between *rad24∆* and *wt.* P values were determined using analysis of variance (ANOVA) with SPSS statistics package 22.0, followed by Games-Howell post-hoc test. We also removed the double mutant *sts5∆ rad24∆*, because as stated by the reviewers, it wasn’t adding to the point of the figure.